# 🦕 REXBENCH: CAN CODING AGENTS AUTONOMOUSLY IMPLEMENT AI RESEARCH EXTENSIONS?

## ABSTRACT

Agents based on Large Language Models (LLMs) have shown promise for performing sophisticated software engineering tasks autonomously. In addition, there has been progress towards developing agents that can perform parts of the research pipeline in machine learning and the natural sciences. We argue that research *extension* and its implementation is a critical capability for such systems, and introduce **REXBENCH** to support the evaluation of this capability. **REXBENCH** is a benchmark consisting of realistic extensions of 12 research papers that aim to investigate *novel* research hypotheses. Each task is set up as an extension to an existing research paper and codebase, accompanied by domain expert-written instructions. **REXBENCH** is robust to data contamination, and supports an automatic evaluation infrastructure that executes agent outputs to determine whether the success criteria are met. We use this benchmark to evaluate 13 LLM agents implemented using three different frameworks: aider, Claude Code, and OpenHands. We find that all agents fail to autonomously implement the majority of the extensions, with the best agent at around 31% success rate. Although the success rate improves with additional human-written hints, the best performance under this setting remains below 48%. This indicates that current agents are still short of being able to handle realistic research extension tasks without substantial human guidance. Based on analyses of prominent failure modes, we put forward actionable short- and long-horizon recommendations for future research coding agent development.

## 1 INTRODUCTION

Interesting research necessarily builds on other research. In this regard, *extensions* of existing research are important starting points to new investigations, potentially building up towards exciting novel discoveries. In light of recent growing interest in building LLM agents that can conduct scientific research in an autonomous manner, we propose **REXBENCH**, a benchmark aiming to evaluate LLM agents' ability to extend existing AI research, with an initial focus on Natural Language Processing (NLP) and Machine Learning (ML). More specifically, **REXBENCH** tests whether LLM agents can autonomously implement research extension experiments via code in a hypothesis-guided manner (Luo et al., 2025), where the extension hypotheses are provided to the system as verbal instructions along with relevant background material including the research paper(s) and the corresponding codebase. Our benchmark consists of realistic extensions of 12 recently published research papers in the field, accompanied by domain expert-written extension instructions (See Appendix C for a sample task instruction). The extension tasks cover various aspects of implementation involving changes to the model, algorithm, data, and evaluation method. The main metric of success is numerical replication of the outcome of domain-expert implemented "gold" solutions for the extension task. We provide an automatic evaluation infrastructure to execute the LLM agent-implemented solutions and evaluate the outcomes. The executions of both the gold solutions and system solutions are conducted in virtual machines with exactly the same specifications to control for experimental variation. **REXBENCH** furthermore is robust to data contamination issues that affect the majority of existing benchmarks: the solutions and the success criteria for our extension tasks only exist in our held-out evaluation infrastructure and do not exist anywhere online.

We tested thirteen agents based on an array of Large Language Model (LLM) backbones (Claude 4/3.7 Sonnet (Anthropic, 2025; 2024), GPT-5 (OpenAI, 2025), o1 (Jaech et al., 2024), o4-mini, and DeepSeek R1 (Guo et al., 2025)), using three different agent frameworks (aider, Claude Code,

Figure 1: End-to-end workflow of **RExBench**: (1) An LLM agent receives inputs consisting of the research paper(s), the original codebase, and an extension instruction; (2) the system implements the extension and a patch file is obtained; (3) the patch is applied to the original code and executed via our evaluation infrastructure; and (4) the results are evaluated using specified metrics.

OpenHands). Many agents struggled on our benchmark, achieving success rates close to zero for most tasks. Agents with Claude 4/3.7 Sonnet and GPT-5 as backbone showed promise, often showing qualitative signs of success even when they did not achieve final success. Nevertheless, even the best-performing agents succeeded less than one third of the time on average (31% success rate for OpenHands + {Claude 4 Sonnet, GPT-5}), leaving much headroom for progress.

While the current **RExBench** tasks pose substantial challenges for the agents tested, most extensions do not require major rewriting of the codebase and are not extremely challenging in terms of complexity (at least to a PhD-level domain expert). We thus consider the release of this specific set of tasks and the paper as a contribution about the broader framework for evaluating research extensions (and the opportunities it may bring), which will motivate the development of more challenging extensions covering broader scientific domains, inviting contributions from the community.

## 2 RELATED WORK

Recent advancements in LLMs and agentic frameworks motivated discussions about their applicability to scientific research. This includes using LLMs and LLM-based agents for research automation (Li et al., 2025; Skarlinski et al., 2024; Jansen et al., 2025; Ziems et al., 2024; Choi, 2024; Boiko et al., 2023; Gottweis et al., 2025; Kitano, 2021; Gandhi et al., 2025) and benchmarking their ability to conduct research in the domains of social sciences, statistics, and natural sciences (Tian et al., 2024a; Chen et al., 2024; Laurent et al., 2024). For ML research, current attempts span automation across all stages of the research process: from ideation (Si et al., 2024) to experiment design (Abramovich & Chechik, 2025) and execution (Siegel et al., 2024; Xiang et al., 2025), to paper review and meta review (Du et al., 2024). There have also been early attempts to automate the full research pipeline (Lu et al., 2024; Kon et al., 2025).

Another line of work benchmarks coding and software engineering skills. Specific skills targeted include resolving GitHub issues (SWE-Bench, Jimenez et al., 2024, SWE-rebench, Badertdinov et al., 2025), debugging LeetCode problems (DebugBench, Tian et al., 2024b), resolving configuration/dependency issues in research environment setups (SUPER, Bogin et al., 2024), and solving tasks in a terminal environment (The Terminal-Bench Team, 2025). In a similar vein, other benchmarks assess more comprehensive ML problem-solving and code implementation skills. MLE-bench (Shern et al., 2024) and DSBench (Jing et al., 2024) design machine learning and data science tasks akin to Kaggle-style competitions; MLAgentBench (Huang et al., 2024) gathers classical ML tasks such as regression and model training problems as well as Kaggle challenges; DataSciBench evaluates data analysis and visualization skills with novel evaluation pipelines (Zhang et al., 2025); and ML-Dev-Bench (Padigela et al., 2025) focuses on the full ML development workflow.

The most directly relevant efforts to ours are benchmarks that evaluate ML problem-solving and software engineering capabilities in research settings. Curie (Kon et al., 2025) aims to evaluate the ability to plan and execute experiments; BLADE (Gu et al., 2024) is designed to automatically evaluate agents' approaches to open-ended data-driven research questions. Paper2Code (Seo et al., 2025) introduces a multi-agent LLM framework to translate ML papers into codebases through a stage-wise design, and PaperBench (Starace et al., 2025) evaluates research agents using a compilation of

coding tasks targeting replication of 20 ICML papers. **REXBENCH** has a similar goal as PaperBench and, to some extent, Curie, in benchmarking of ML and AI research code generation. However, a key distinction is that instead of evaluating replications (PaperBench) or very general questions that can often also be answered without running experiments (Curie), we focus on *novel research extensions*. Thus, **REXBENCH** is able to evaluate agent performance on previously unseen/unimplemented research hypotheses which greatly alleviates data contamination concerns.

# 3 BENCHMARK DESIGN

## 3.1 RESEARCH EXTENSION TASK

**Task** We define our research extension task as a code implementation problem, where the input consists of an existing research paper, an accompanying codebase, and an instruction that verbally describes an extension proposal and how this should be tested. An example of a simple extension is: "What would happen if the same experiment in paper X used an open-source model like Llama 3 70B instead of GPT-4o?" (see Appendix C for an actual example). Given this input, a system must produce as output edits to the input codebase that implements the extension proposal.

**Desiderata** The core aim of our benchmark is to *automatically* assess how well an agent can *autonomously* implement *realistic* research extensions. These goals are to some extent in conflict with each other: Realistic research extensions tend to be quite open-ended, which makes automatic assessment challenging or impossible; limiting tasks to the availability of simple automatic measures, on the other hand, may constrain the task too much for it to be still realistic. We strike a balance between these two goals by using automatic tests that allow the agent to tackle the task through any means, as long as this leads to results comparable to the ones from our gold implementation. The task setting of requiring implementation on top of an existing codebase and evaluation through controlled execution environments (random seed, hardware, packages, etc.) serves to improve the reliability of the numeric output-based automatic evaluation. Nevertheless, each extension proposal included in the benchmark still cannot be too open-ended or exploratory, and therefore consist of specifically-scoped questions that can have well-defined numeric targets. To ensure that agents autonomously implement extensions, the granularity of our instructions are calibrated at a level that still requires the model to thoroughly analyze the codebase and form its own plan for the extension. Furthermore, at no point of the evaluation do humans provide additional supervision. Finally, one of the biggest challenges with LLM evaluation is data contamination. If solutions to any of the tasks are openly available on the web, LLMs that serve as the backbone for the agents may have been trained on the solutions (also noted as a possible issue in PaperBench (Starace et al., 2025)), rendering it impossible to establish whether success stems from memorization or autonomously solving the task. We circumvent this problem by including only novel research extensions, either in terms of the idea itself or implementation. To the best of our knowledge, none of our extensions exist on top of the existing codebases publicly; we store all the gold extensions in private Bitbucket repositories.[1] Furthermore, our privately hosted evaluation infrastructure prevents models from accessing the evaluation scripts or reference solutions.

## 3.2 BENCHMARK COMPOSITION

Our benchmark consists of research extensions building upon papers and codebases primarily in the NLP and broader AI domains, taking into consideration the availability of expertise within the team as well as the availability/replicability of the code released. The full list of papers is in Table 1.[2] The specific extension proposals were selected to span various dimensions of change including changes to the model, dataset, algorithm, and evaluation. In addition to this consideration, we imposed the following constraints on the extension proposals for scientific rigor and feasibility of the experiments: (1) Important empirical trends from the original paper relevant to the extension proposal must replicate; (2) the gold implementation of the extension proposal must replicate (e.g., if the gold implementation requires making calls to a closed API-based model, this may not replicate

---

[1] We use Bitbucket instead of GitHub since GitHub data has been used in the past to train LLMs and it is unclear whether this may also be true for some private repositories.

[2] Two of the tasks (COGS, Othello) involve implementing an extension proposal from another paper on top of the codebase of the original paper, where the implementation of the extension is either not publicly available or is not implemented as an edit of the original codebase. For these tasks, there are two relevant papers.

in the future due to model deprecation); and (3) the estimated runtime of each gold implementation should be shorter than 12 hours on a single A100 GPU. The final dataset includes the extension instruction, target research papers in both .pdf and .md format (converted using `PyMuPDF4LLM` to accommodate agents that lack the ability to read .pdf files), and the original codebase.

## 3.3 BENCHMARK CONSTRUCTION PROCESS

For each extension proposal, a domain expert (PhD student-level or above) first verified that the original codebase replicates the results of the associated paper on our virtual machines (details to follow). Then, they implemented the "gold" edits for the target extension and recorded the numerical outcomes, ensuring that the runtime does not exceed 12 hours. This implementation process and the outcomes were validated by at least one other author. Finally, the domain expert wrote the instruction that consists of a brief description of the original paper, the extension proposal, and how this proposal should be tested (see Appendix C for a full instruction for one of our tasks, WinoDict). The description of the "how" was deliberately high-level to meet the desideratum of evaluating a sufficiently autonomous capacity. Nevertheless, since the instructions should not be confusing or ambiguous, they were polished through multiple rounds of revisions by multiple authors to improve clarity. Importantly, if we foresaw degrees of implementation freedom that may introduce random variation, we controlled for this by specifying constraints (e.g., use an implementation of Pearson correlation function from the `scipy` package as opposed to implementing this from scratch). During this revision process we furthermore ensured that each extension was self-contained. No part of the gold edits required information external to the set of inputs provided to the system. As a part of the revisions for self-containment, we provided information such as specific model identifiers and explanations of necessary hyperparameters not in any README or the paper as a part of the instruction, and added version information for all of the packages (via an `environment.yml` file).

## 3.4 EVALUATION METRICS

Our main metric is final success rate, which measures whether the outcome of executing the model-implemented code falls within the target range (more details below). We define two additional metrics for finer-grained analyses: execution success rate and file recall. We describe each metric below.

**Final Success Rate**  Final success rate evaluates whether the model correctly implements the specified research extension. This evaluation either checks whether the final results exactly match the results of the reference implementation (if the run is fully deterministic) or it checks whether the results fall within a *gold range* that we obtained by running the gold implementation multiple times with different random seeds to account for output variability. In the latter case, a model solution is considered successful if its final execution outcome falls within this bound.

**Execution Success Rate**  Execution success rate checks whether the generated code runs without errors in our evaluation environment. This metric evaluates the general well-formedness of the code and contextual understanding sufficient to avoid runtime errors.

**File Recall**  File recall quantifies whether files edited in the gold solution were also edited by the model: File Recall $= |\text{Files}_{\text{agent}} \cap \text{Files}_{\text{gold}}| / |\text{Files}_{\text{gold}}|$. The limitation of this measure is the dependency on the gold solution. Technically, a solution could achieve zero file recall with perfect final success. E.g., if a model solution was exactly equivalent to gold but created new files with identical content instead of editing, and changed references appropriately in the repository, this would be the case. Still, we take human expert edits to reflect a reasonably efficient set of modifications.

## 3.5 EVALUATION INFRASTRUCTURE

**Submission format**  Our metrics defined above require execution of agent generated code. We propose to conduct this execution on a virtual machine to control for hardware specification and package dependencies. We will host this infrastructure using our own resources, and conduct evaluation asynchronously at a regular interval to update the leaderboard with the submissions we receive, similarly to Jimenez et al. (2024). The submissions will be received in the form of git patch

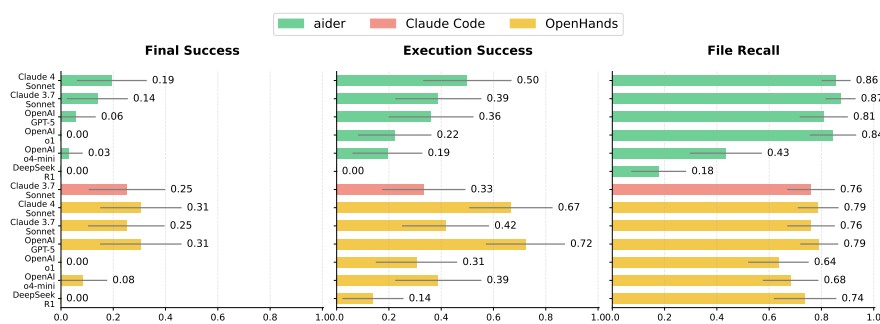

Figure 2: Agent performance on **REXBENCH**. The color coding indicates the agent framework and the y axis indicates the the backbone LLM. Results include three runs per task to account for agent random variation. Error bars show standard error of the mean of all runs per model computed using the closed form formula ($2\sigma$, no normality assumption).

files (as opposed to full edited repositories) to streamline the submission process. Additionally, we will request agent log files to verify that the task was completed autonomously by an agent.

**Infrastructure pipeline** We host our evaluation infrastructure based on the OpenStack platform on an academic cloud computing service. For each patch file received, we launch an Ubuntu virtual machine instance with a 20GB root disk, where we run a task-specific Apptainer container (Singularity, 2021) that has the original codebase and evaluation scripts pre-loaded and the environment set up. Each instance is also equipped with task-specific hardware: either a single NVIDIA A100 40GB GPU, a single K80 12GB GPU, or just a CPU (see Appendix B, Table 2). To control for random variation of the execution outcomes to the best of our effort, we (1) fix all random seeds in the codebase wherever possible, and (2) run the evaluations with exactly the same hardware configuration as our gold runs. Inside the container, we apply the patch file and execute a single bash script `run_apptainer.sh` that contains all necessary commands (this requirement is also provided in the task instructions). We limit the runtime to 12 hours, which is around twice the duration of the gold solution with the longest runtime among our extension tasks (see Table 2 for all estimated runtimes). Once task execution is complete or the attempt crashes, any result files and task execution logs are copied to an external storage volume. We then delete the virtual machine instance and evaluate the results. This setup ensures a fully containerized and task-level parallelizable evaluation infrastructure.

## 4 EXPERIMENTS

### 4.1 MAIN EXPERIMENT

We follow steps shown in Figure 1 and evaluate thirteen LLM agents, combining three agent frameworks with various LLM backbones (discussed below). We pass the full set of inputs for each task one by one to the agent to evaluate each task independently of each other. We run each task three times with the same agent model to account for agent random variation.

### 4.1.1 BASELINE AGENT DESIGN

We used three different agent frameworks: two open-source (aider, aider AI (2023) and OpenHands, Wang et al. (2025)) that we adapted for the task, as well as the proprietary Claude Code. aider and OpenHands both support several backbone LLMs. We evaluated `GPT-5`, `o1` and `o4-mini` (OpenAI), `Claude 3.7 Sonnet` and `Claude 4 Sonnet` (Anthropic), and the open-weight `DeepSeek-R1`. We discuss a few design decisions shared between our agents below. Note that this does not imply future submissions to our benchmark should be subject to the same design decisions.

**Shared design considerations** For better runtime controllability, we disabled Python code execution for all agents to the best of our effort. Regarding the settings of the backbone LLMs, we set the temperature to 0.7 for Claude 4/3.7 Sonnet and DeepSeek-R1. For GPT-5, o1, and o4-mini, we used

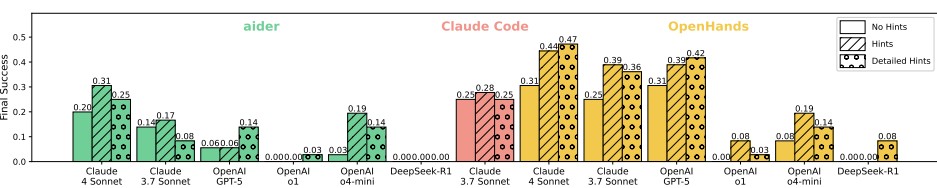

Figure 3: Final success rates for each agent-LLM combination and hint level.

the default settings, as these models do not support custom temperature adjustment. We specified the reasoning effort as "medium" for all OpenAI models. As discussed in Section 3.5, our evaluation infrastructure requires git patch files. We created the patch files using a separate script after the agents had made changes to the codebase. We discuss individual implementation details in Appendix D.

## 4.2 EXPERIMENT WITH ADDITIONAL HINTS

We conduct an additional set of experiments where we provide different levels of hints to the agents. This experiment serves two purposes: (1) as a layer of sanity check that our tasks are possible to solve; (2) to gain a more fine-grained understanding of where the difficulties lie, if the agents do find the tasks difficult without hints. We design two levels of hints, where the first level of hints mainly provides help with information localization, and the second level of hints provides a step-by-step implementation guidance. Information localization hints, for instance, help find specific locations of edits by directly naming a file to be edited ("You would need to edit `test_function()` in `src/testfile.py`"), help find necessary information ("Look at the README to find the descriptions of the hyperparameters"), or directly provide certain pieces of information that are part of the given input but nontrivial to find ("Use ID #1014 for the special token"). On the other hand, the second level of hints breaks down the gold solution into concrete implementation steps. Therefore, we expect the second level of hints to yield substantially higher success rates. In our experiments, hints are cumulative; when providing the second level of hints, the first level of hints is also provided.

## 4.3 RESULTS

**Main experiment** Figure 2 shows our main results. Most agents struggle with the task, with the best performing agents (OpenHands + {Claude 4 Sonnet, GPT-5}) achieving 31% average final success rates. Claude 4 Sonnet and GPT-5 were the best backbone LLMs—when different LLMs were available, they yielded the strongest performance. All agents achieved nonzero execution success rates except for DeepSeek-R1, which failed completely. Claude 4 Sonnet and GPT-5 again performed best, with success rates of 67% and 72% respectively when combined with OpenHands. The agents overall achieved high file recall, showing that they were able to locate core edit targets based on the instructions.

**Additional hints** Figure 3 (and Table 4 in Appendix F) show the results of additional experiments with two different hint levels. Generally, hints improve the final success rate, but tend to help less when the default success rate was zero, suggesting there is a base level of competence required to make use of the hints provided. With the hints, we could boost the performance of the best agents, OpenHands + {Claude 4 Sonnet, GPT-5}, achieving 47% and 42% final success rates, respectively.

## 4.4 RESOURCE CONSUMPTION

Based on the final success rate, we plot the cost/time vs. performance tradeoff (Figure 4), showing that aider + o4-mini, aider + Claude 4 Sonnet and OpenHands + Claude 3.7 Sonnet lie on the Pareto frontier for both cost and time. We provide the full time and cost estimates for agent runs in Appendix F, Table 6. In terms of token usage statistics, aider consistently used 2 turns due to its non-iterative design. Claude Code used 25–35 turns and OpenHands used 17–93 turns, making use of active multi-turn structures. Due to its closed-source nature, we could not obtain token counts

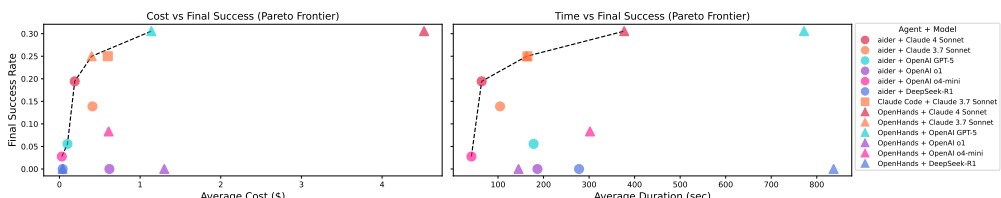

Figure 4: Cost effectiveness and time efficiency of coding agents on REXBENCH.

for Claude Code.[3] OpenHands used the most tokens, especially with Claude 4 Sonnet and GPT-5, reaching up to 1.86M prompt tokens (almost 560 times more than aider). As the hint levels increased, both turns and token usage in OpenHands tended to grow, while the turns in Claude Code decreased. See Table 5 in Appendix F for token usage statistics by model and by hint levels.

## 5 ANALYSIS AND DISCUSSION

### 5.1 PATTERNS OF ERROR

We discuss notable error patterns, dividing them into explicit and implicit errors. We treat cases where the agent-generated code failed to execute as explicit errors, and cases where the execution succeeded but the experimental outcome did not match the numerical criteria as implicit.

**Explicit errors** Explicit errors were automatically identifiable from execution logs. The most common source of error was Python value errors (e.g., incorrect chat templates or invalid parameters). These errors were observed in all agents. Another common source of error was empty patch files due to the failure of the agent to modify any code. The majority of the empty patch file errors were from aider + {DeepSeek, o4-mini}. We attribute this to the non-iterative nature of this agent framework: agents need to solve the entire extension task in one shot rather than breaking it down, often leading to incomplete or failed command executions during agent runs. Beyond these cases, most explicit errors were Python errors and they were mostly Python native errors rather than library-specific errors. Agents with Claude or GPT-5 as backbone led to fewer SyntaxErrors (in particular, OpenHands + {Claude 4/3.7 Sonnet, GPT-5} had no SyntaxErrors), whereas o1 produced SyntaxErrors frequently. There were also several cases of execution timeout, which occurs when the experiment runtime exceeds the limit of 12 hours we set (no gold solution required more than 6 hours). The full error distribution is shown in Figure 7 and Tables 7 and 8 in Appendix F.

**Implicit errors** Analysis of implicit errors (execution success but mismatch with gold outcome) involved greater manual effort because it required a holistic review of agent edits. Therefore, we focused our analysis on the top 2 agents (OpenHands + {Claude 4, GPT-5}). Overall, the agents' implicit errors were categorizable into errors in implementation logic and errors in value (e.g., within-bounds index errors, incorrect hyperparameters or paths)—the ratio of logic vs. value errors was about 2:1 for Claude 4 and about 1:1 for GPT-5. We also estimated the debugging difficulty from the manually identified sources of error, using the scale of easy (requires small local fix), medium (requires logical but local revisions), and hard (requires holistic revisions). For both models, the majority of the errors were easy to debug. OpenHands + GPT-5 had more implicit errors, especially ones falling into the easy and medium categories (21 easy, 12 medium, 3 hard) compared to OpenHands + Claude 4 (16 easy, 4 medium, 4 hard), revealing a qualitative difference in the agents' solutions although the quantitative success rates were similar. Many of the medium and hard implicit errors arose from agents "over-editing" the code beyond the given instructions (e.g., adding extra (incorrect) exception handling or changing irrelevant flags/prompts). These unrequested edits often caused silent failures or subtle deviations from the gold implementation leading to markedly different results, making debugging harder. Task-specific examples are discussed in Appendix G.

---

[3]As of now, token counts for Claude Code have become available, but this feature was not available when our experiments were conducted.

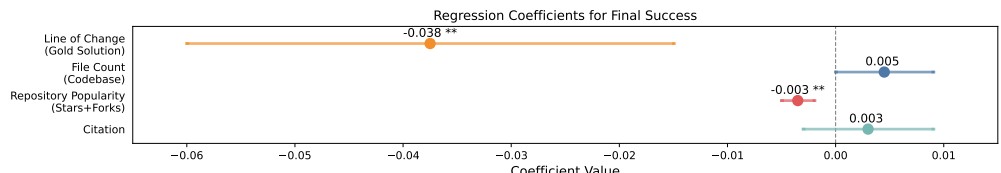

Figure 5: Regression coefficients with 95% confidence intervals for predictors of final success. (Regression model: `final_success ~ line_change + file_count + repository_popularity + citation + (1 | model)`). (*: $p < .05$, **: $p < .01$)

### 5.1.1 QUALITATIVE OBSERVATIONS

**Implicit errors increase as model capacity increases, but are more difficult to analyze**  A high-level observation is a general pitfall associated with stronger models (for our task and coding tasks more generally): the cause of failure is difficult to identify. Better models produced more implicit errors (e.g., OpenHands + Claude 3.7: 6, OpenHands + Claude 4: 24), where the code successfully executes but the outcome is incorrect. In such cases, the reasons behind failure were not always easily traceable even for the experts who implemented the solutions. This highlights the need for heavy sanity checks (perhaps supported by system design) if an agent were to be deployed in practice. Plausible-looking implementations that execute can lead researchers to draw conclusions from faulty implementations, and over-reliance on coding agents may lead to a proliferation of incorrect results.

**Overthinking is often an issue**  A prominent issue with weaker LLMs (Deepseek-R1, o1, o4-mini) was overthinking, where the thinking process was excessive both in terms of the number of output tokens and agent runtime, frequently leading to no actual output in terms of code generation. aider + DeepSeek-R1 was especially prone to this behavior, overthinking being one of the most prominent failure modes (close to one third of total failures). One possibility is that models' reasoning behavior somehow clashes with the reasoning/"thinking" loop of the agent framework, but this pattern appears weaker in Claude 4/3.7 Sonnet and GPT-5, which are themselves reasoning models.

**Agents vary in their ability to make use of hints**  As noted in Section 4.3, providing additional hints did not always improve agents' success rates, nor did more hints necessarily yield greater gains. While the best agents (OpenHands + {Claude 4, GPT-5}) benefited from both levels of hints, many others showed no improvement. The best agents benefitted more from the second level of hints, suggesting that a certain baseline competence or underlying model capacity may be required to leverage more detailed, human-written guidance. We observed idiosyncratic task-level variation as well; for instance, for the Othello task, OpenHands + Claude 3.7 Sonnet and Claude Code achieved 100% success rate with no hints and with the first level of hints, but 0% success rate when additionally given the second level of hints. Upon closer observation, these agents employed a qualitatively different strategy with the second level of hints. However, it was not the case that this particular hint was misleading, since the 2 best performing agents were able to use this hint to achieve 100% success rate on this task. This can be interpreted as models varying in their ability to implement different equally plausible solutions, and the step-by-step guideline in the second level of hints specifying a different solution from the one that the model could implement easily. We noticed this pattern for two tasks (Othello and Tree-of-Thoughts), but not in general.

### 5.2 WHAT MAKES AN EXTENSION DIFFICULT FOR AGENTS?

We hypothesize four sources of difficulty that could contribute to agent failure: (1) implementation effort; (2) codebase size; (3) unfamiliarity with the codebase; and (4) unfamiliarity with the research topic. We operationalize them as: (1) lines of code change in our gold solution; (2) file counts of the original codebase; (3) GitHub stars + forks (repository popularity); and (4) Google Scholar citations of the research paper(s), respectively. We use these as predictors of final success in a mixed-effects model with model identity as a random effect. Figure 5 shows the regression coefficients. Lines of code changes has a significant negative effect ($\beta = -0.038$, $p < 0.01$) on final success, indicating that tasks with higher implementation effort are more difficult. Repository popularity had a significant effect but the effect size was negligible. Other factors were not statistically significant.

## 6 CONCLUSION

We presented **REXBENCH**, a benchmark evaluating the autonomous capacity of AI systems to implement hypothesis-driven research extensions in the domain of AI research. **REXBENCH** consists of realistic but well-scoped extension tasks motivated by existing research. To perform well, a system must be able to understand the expert-written extension instructions situated in specific research context, understand the structure and logic of the original codebase, and autonomously plan and implement the requested extension. Our tasks are by design robust to data contamination due to the extensions requiring novel implementations whose solutions are not available publicly. Experiments with various agent frameworks combined with competent backbone LLMs show that most systems struggle on our benchmark, with the best performing models (OpenHands + {Claude 4 Sonnet, GPT-5}) achieving 31% extension success rate. Notably, agents with o1 or DeepSeek-R1 as backbone showed (close to) zero success rate. Nevertheless, closer analysis of the best models revealed promise: the strongest backbone LLMs (Claude 4 Sonnet, GPT-5) achieved higher execution success rates than weaker models,with implementations often syntactically valid and logically on the right track. This observation, taken together with the large headroom, highlights the utility of **REXBENCH** for guiding future developments of research agents. Finally, based on the analyses of the agents tested in this work, we put forward several actionable recommendations for the future.

**Short-horizon recommendations:**

- **Incorporate iterative design**: Our findings show that iterative design is critical for success on our tasks: aider (a single-turn framework) showed weaker performance in general, and many success scenarios for multi-turn agents could be attributed to effective use of the previous turns' output. For instance, in the CheckEval task, OpenHands + GPT-5 used one turn to inspect a file's structure with bash before writing code in the next.
- **Support scratchpads**: Agents frequently failed on the basis of small errors such as path mis-specification. Such errors could be easily caught if agents can make use of a "scratchpad" where small code snippets can be executed.
- **Support "repair" mechanism**: Agents should incorporate a mechanism to repair a step in their action trajectory, for instance by reverting the changes made in the step and re-initiating the LLM call. One use case of this would be detecting and repairing overthinking in the LLM output, which was a prominent failure mode in several agents, especially with DeepSeek-R1 as backbone, that resulted in no code edits.

**Longer-horizon recommendations:**

- **More stringent verification**: One of the most concerning observations from our analysis is the increasing trend of implicit errors as the capacity of the backbone LLM grows. Under a benchmarking setup, numeric mismatches of the outcome to the gold solution easily indicates failure, but in real deployment scenarios, there exist no gold solutions. This indicates a need for more stringent verification processes, ideally by agent design rather than relying on manual verification from the end users.
- **Prevent over-editing**: A prominent failure mode of the strongest agents was "over-editing", where agents make unrequested modifications that often lead to implicit errors. Our findings show that simply instructing an agent to "keep everything else not specified constant" (see Appendix C) is insufficient. A general improvement in hallucination reduction and instruction-following would help, but for research coding where fine-grained controls of experimental details is critical, a more targeted solution for over-editing may be beneficial.
- **Improve handling of long contexts**: Our analysis shows that the most important factor to agent failure is the size of the required edits. Given that the maximum lines of change in the gold solutions in our benchmark is not huge (in the magnitude of hundreds), there is a need for future agents to handle long contexts better, both within and across file boundaries.

**The future of REXBENCH** Finally, as discussed in the introduction, we view the release of **REXBENCH** and this paper as a motivating start to a larger community-driven effort. While our tasks were primarily in the AI domain with a focus on topics aligning with the expertise of our team, we believe the format of the extension task and evaluation framework shown in Figure 1 are broadly applicable beyond our set of tasks. We hope the current set draws community interest in research extensions as an interesting problem for agents, and hope to collaborate with researchers and/or solicit community contributions for a comprehensive coverage of task domains and implementation complexities.

## ETHICS STATEMENT

In this work, we showed that current LLM-based agents cannot reliably produce code for AI research without additional human supervision. We based this argument on the low final success rate of all evaluated agents, as well as the danger of the increasing trend of implicit errors as model capacity improves. Given the rapid progress of AI research and model development, it is a likely possibility that new agents would perform significantly better on this benchmark in the near future. The biggest risk we therefore foresee is that good performance on this benchmark is seen as a sufficient condition for reliable agents rather than a necessary one. While we consider the benchmark to be well-suited for measuring progress in the development of future agents, good performance should NOT be seen as sufficient evidence for an agent being able to autonomously produce reliable research code.

We also would like to highlight again that the baseline agents we evaluated for this work did not reach the level of competence that would translate into autonomous research extension capacities in the real world. Given the difficulty of debugging, deployment of such systems without rigorous verification measures faces the danger of leading researchers to draw conclusions from faulty implementations and of the erosion of trust in published results.

Furthermore, as discussed in Section 3.1, a benchmark being realistic inherently conflicts with the ease of automatic evaluation. In particular, a task like research extension can be extremely open ended in reality, even when constrained with a specific proposal and hypothesis. We opted for a middle ground where we do not enforce strong limits on *how* a system may implement the target extension and condition final success only on alignment of numerical outcomes. This necessitated a stronger control for sources of variation, which led us to write instructions as self-contained and unambiguous as possible. This setting is idealized as the instructions are much more informative and clearer than an actual task a human researcher faces, even in scenarios where the extension idea is provided to them (e.g., an advisor suggesting to a PhD student "How about we try X?"), missing out on the real difficulties lying in the initial trial-and-error concretization step.

Finally, executing machine-written code always bears safety risks and providing AI agents with too much freedom for exploration may enable them to cause harm. To mitigate this risk, we narrowly scoped the implementation tasks in our experiments fully based on human-generated hypotheses and instructions. Furthermore, any machine-written code was executed in a containerized environment without internet access. We recommend similar setups for the execution of any code that is output by AI agents.

## REPRODUCIBILITY STATEMENT

Our dataset and the code for our baseline agents are submitted as supplementary material. We have furthermore taken the following steps to ensure reproducibility of our experimental results. First, we fix random seed values across multiple libraries, including Python's built-in random module, NumPy, PyTorch (CPU and CUDA), and CUDA cuDNN, in order to control for nondeterminism in obtaining the execution outcomes from the gold solutions. Second, we execute both the gold solutions and agent solutions within virtual machines using Apptainer containers. These containers are configured with identical hardware resources and software/library versions, ensuring that all experiments run under consistent conditions and that reported performance is not affected by hardware or software variability. In addition, while we refrain from releasing gold solutions publicly due to contamination concerns, we welcome interested researchers to request access to specific ground-truth solutions for further validation and standardization purposes.

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

## A  LIST OF PAPERS FOR REXBENCH EXTENSION

Table 1: List of papers that form the bases for extensions in **REXBENCH**.

| Identifier | Extension Type | Source paper | Venue |
|---|---|---|---|
| CheckEval | Evaluation | *CheckEval: A reliable LLM-as-a-Judge framework for evaluating text generation using checklists Lee et al. (2025)* | EMNLP 2025 |
| COGS | Model | *COGS: A Compositional Generalization Challenge Based on Semantic Interpretation Kim & Linzen (2020)* | EMNLP 2020 |
| | | *The Devil is in the Detail: Simple Tricks Improve Systematic Generalization of Transformers Csordás et al. (2021)* | EMNLP 2021 |
| Entity Tracking | Model | *Code Pretraining Improves Entity Tracking Abilities of Language Models Kim et al. (2024)* | Preprint |
| Explain then Translate | Algorithm | *Explain-then-Translate: An Analysis on Improving Program Translation with Self-generated Explanations Tang et al. (2023)* | EMNLP Findings 2023 |
| Instruction Tuning | Model | *Instruction Following without Instruction Tuning Hewitt et al. (2024)* | Preprint |
| Mission Impossible | Data/Evaluation | *Mission: Impossible Language Models Kallini et al. (2024)* | ACL 2024 |
| Othello | Data/Evaluation | *Emergent World Representations: Exploring a Sequence Model Trained on a Synthetic Task Li et al. (2023)* | ICLR 2023 |
| | | *Emergent Linear Representations in World Models of Self-Supervised Sequence Models Nanda et al. (2023)* | BlackboxNLP 2023 |
| Reasoning or Reciting | Model | *Reasoning or Reciting? Exploring the Capabilities and Limitations of Language Models Through Counterfactual Tasks Wu et al. (2024)* | NAACL 2024 |
| Re-reading | Algorithm | *Re-Reading Improves Reasoning in Large Language Models Xu et al. (2024)* | EMNLP 2024 |
| Tree of Thoughts | Algorithm | *Tree of Thoughts: Deliberate Problem Solving with Large Language Models Yao et al. (2023)* | NeurIPS 2023 |
| VariErr-NLI | Model/Data | *VARIERR NLI: Separating Annotation Error from Human Label Variation Weber-Genzel et al. (2024)* | ACL 2024 |
| WinoDict | Data/Evaluation | *WinoDict: Probing language models for in-context word acquisition Eisenschlos et al. (2023)* | EACL 2023 |

Table 1 shows the papers forming the basis of the RExBench extensions.

## B  DETAILED EXPERIMENTAL SETUP

Table 2: Resource requirements for each task.

| Task | Instance Type | Runtime Duration (Gold Solution) |
|---|---|---|
| CheckEval | CPU | 1m |
| COGS | K80 | 5h |
| Entity Tracking | A100 | 2h |
| Explain then Translate | CPU | <1m |
| Instruction Tuning | A100 | 5h |
| Mission Impossible | A100 | 4h |
| Othello | K80 | 1h |
| Reasoning or Reciting | A100 | 6h |
| Re-reading | A100 | 30m |
| Tree of Thoughts | A100 | 20m |
| VariErr-NLI | A100 | 10m |
| WinoDict | A100 | 30m |

Table 2 shows the details about the execution environment for each task.

## C  AN EXAMPLE TASK INSTRUCTION (EXTENSION OF WINODICT)

---

**WinoDict Task Instruction**

### PROBLEM DESCRIPTION

#### BACKGROUND

The paper *WinoDict: Probing language models for in-context word acquisition* (Eisenschlos et al. 2023) attempts to measure LLMs' ability to learn novel words during inference. They rewrite Winograd-style co-reference resolution problems by replacing the key concept word with a synthetic but plausible English word and adding the definition of the new concept as a suffix. Building on this work, we would like to further consider a learning setting where the form of the learned words coincides with existing English words and explore how their existing meanings may interfere with the models' word acquisition from the given definition. The hypothesis is that overriding existing words would be more difficult, and the frequency of the existing words may also modulate the effect.

The paper will be available inside the provided repository in both PDF format as `eisenschlos_et_al_2023.pdf` and markdown format as `eisenschlos_et_al_2023.md` if you need to refer to it.

#### EXTENSION TO BE IMPLEMENTED

Your task is to modify the codebase provided to generate new Winodict datasets by replacing the target word being learned with an existing English word. The new dataset should be stored under the directory `./data`. Your replacement should consider the POS tags of the original word - they should be matched. We will only consider four POS categories for word replacement: nouns, verbs, adjectives, and adverbs. To test the possible effect of frequency, sample the candidates from different frequency groups:

1. Top Group:

   - Verbs, Nouns, Adverbs: Select the top 20% most frequent words
   - Adjectives: Select the top 35% most frequent adjectives (to match the sample set size)

2. Bottom Group:

   - Verbs, Nouns, Adverbs: Select the bottom 20% least frequent words
   - Adjectives: Select the bottom 35% least frequent adjectives.

3. All Group:

   - Verbs, Nouns, Adjectives, Adverbs: Include all words, no frequency-based filtering

Assume that the frequency information will be provided in a form of four files corresponding to each POS, named `1_all_rank_noun.txt`, `2_all_rank_verb.txt`, `3_all_rank_adjective.txt`, `4_all_rank_adverb.txt`, under the directory `./words/`. Each file lists words in descending order of frequency from the British National Corpus. To generate the dataset, you need to create word candidates based on the files and sample words from those candidates.

From each group, sample words from the candidate lists to generate the new Winodict dataset. Ensure that the replacement word is inflected to match the morphological properties of the original word being replaced. For instance, if the original word is a past tense verb, the selected replacement must also be in the past tense. Please use `spaCy` with the `lemminflect` module to inflect the selected words as necessary.

Using the new dataset, you should run experiments on the Winodict-Winograd dataset under the 5-shot setting. Assume that the model can be found under `/stage/hf_cache/gemma-2-9b`, although this may not be visible in the provided repository right now. Furthermore, we will only consider the setting where definitions are appended as suffixes, which are represented as the `last_def` template in the codebase. Save your results as three different files under `./results/`, corresponding to the three sampling groups defined above. They should be named `res_top.json`, `res_bottom.json`, and `res_all.json`.

Please make the experiment runnable by implementing a single script called `run_final.sh` in the root of the repository. This script should call all necessary commands with all parameters specified and should not have any command line arguments itself. It should handle both the dataset generation as well as the execution of the experiments on this new dataset.

Try your best to keep everything else not specified above constant in the original repository. Also, the environment is already set up, so you do not need to install any dependencies or download any more datasets/models. Please refer to `environment.yml` in the repository for an overview of the installed libraries and their versions.

#### EVALUATION

To evaluate the extension, we will execute the `run_final.sh` script you wrote. We will use the three `.json` files mentioned above that contain the final results to evaluate the experimental outcome.

---

# D   DETAILED AGENT CONFIGURATIONS

Table 3: Agent components

| Component | aider | Claude Code | OpenHands |
|---|---|---|---|
| Repo navigation | × | ✓ | ✓ |
| Tool use | × | ✓ | ✓ |
| Bash execution | × | ✓ | ✓ |
| Python execution | × | ✓ | × |

Table 3 provides an overview of what kind of abilities each agent has.

**aider**   aider is an open-source agent framework. We implemented our most basic agent based on aider, using the "diff" edit format where the LLM specifies file changes as search/replace blocks. We allowed up to 5 retries to handle API-side overload errors. Since aider lacks built-in file search capabilities, we added a preliminary stage where the LLM is given the codebase's directory tree along with the task instruction to identify files requiring modification. Unlike Claude Code and OpenHands, our aider implementation does not use bash execution or tools.

**Claude Code**   Claude Code is an interactive command-line tool that can be run from within a repository. We used Claude 3.7 Sonnet as the backbone in our experiments. Claude Code can then navigate the repository and make edits to files or create new files. We manually evaluated Claude Code with the prompt "Read the instructions in instructions.md and carry out the specified task." Furthermore, to avoid Python executions by the agent, we added the instruction "Please do not execute any code, just read relevant files and make any necessary modifications." to the task-specific instructions. Since this tool does not allow for modification of system prompts or support any other customization, everything else about this agent was left as defined by its developers. We additionally applied minimal postprocessing to patches (only needed for the Re-reading task) containing an absolute filepath—the agent was evaluated locally and receives the absolute path to the codebase directory as input in an uncontrollable fashion, but patches are evaluated inside a virtual machine with a different filepath structure.

**OpenHands**   OpenHands is an open-source agent framework that uses an LLM to control a range of pre-defined tools for understanding and modifying codebases. To make this agent more fairly comparable to Claude Code, we modified the system prompt and the agent to disable execution of Python code. The agent was allowed to execute bash commands such as `grep` and `cat`, browse the web, load PDFs in a browser (if compatible with the backbone LLM), and edit files. We prompted this agent with the same one-line instruction as for Claude Code. We evaluated this agent in "headless" mode in which the agent executes the task without any user input until the LLM signals task completion to the agent, or the agent detects a loop or reaches a maximum number of steps (250). As with Claude Code, we applied postprocessing to absolute filepaths to make them compatible with the virtual machine evaluation environment, since the OpenHands agent is run inside its own Docker container.

# E   TOOL USE/ACTION DISTRIBUTION

OpenHands agents interact with external tools during execution, and we analyze how their tool usage varies across different LLMs. Claude 4 Sonnet and OpenAI GPT-5 showed the highest overall tool usage. File operations (`str_replace_editor`) and bash commands (`execute_bash`) were the most frequently used tools across all models (see Figure 6) but occasionally the agent did also perform web searches or use a browser to render the paper PDF.

# F   DETAILED EXPERIMENTAL RESULTS

Table 4 shows the detailed results for all metrics and each agent-LLM combination across all three hint levels.

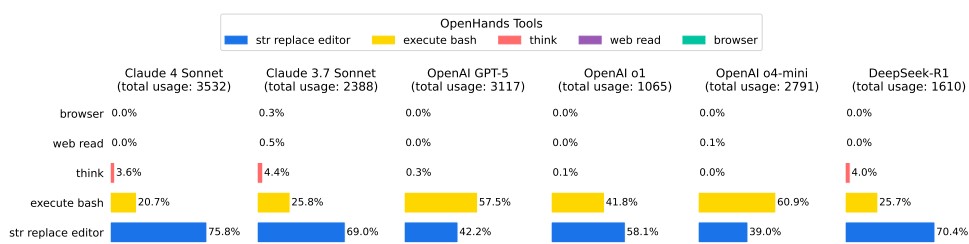

Figure 6: Tool usage distribution across OpenHands agent implementations. Percentages indicate the frequency of each tool type, while the total usage count is shown in each column header.

Table 5 shows the number of turns as well as the number of input and output tokens, averaged across the three runs for each agent.

Table 6 shows the costs and duration for running each agent on a single task on average, as well as the total cost and total durations, based on the main experiment only (providing no hints). Including preliminary and failed runs not reported in the main paper, we estimate that the total compute required for the full project was approximately 4–5x the reported amount.

Table 7 and Table 8 shows the detailed breakdown of errors for each agent and LLM combination.

Tables 9 to 21 show the detailed breakdown of task specific performance for each agent and LLM combination.

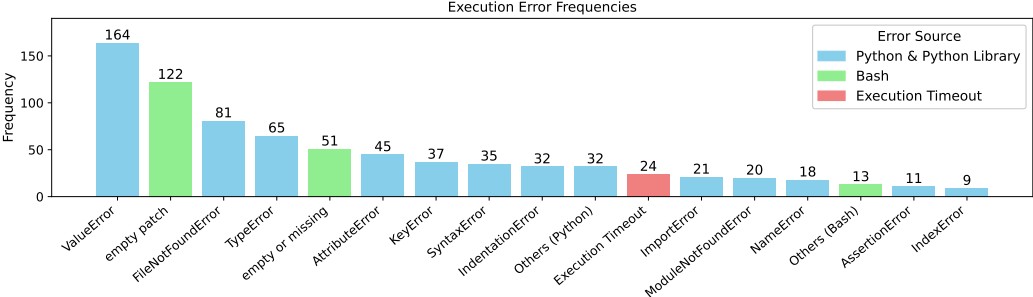

Figure 7: Distribution of execution errors across Python, Bash, and timeout categories. Errors with fewer than 5 occurrences are grouped as 'Others'.

## G  ADDITIONAL QUALITATIVE OBSERVATIONS

**Some agent edits have no practical effect**   Although stronger agents more often write executable code, sometimes the actual implementation has no effect on the output. For example, in the Mission Impossible task, both OpenHands + {Claude 4 Sonnet, GPT-5} incorrectly used the `ParentedTree` class in the `nltk` library. While this code raised a `ValueError`, the agent's implementation used a `try-except` block, returning the parse tree from the original paper as a fallback value, meaning the script still executed. In another instance, OpenHands + GPT-5 incorrectly tried to access the content returned by a method in the `radon` library. The code logic meant that if no function was found using this method, a default value of 0.0 was returned. This in effect meant that the final numerical results were identical to the original paper's experiments. These observations reaffirm the importance of rigorous verification before deploying these systems in the real world.

**Detailed observations on over-editing**   As mentioned in Section 5.1, despite the constraints specified in our task instructions for the agents, we occassionally observed agents making unnecessary additional code edits. For example, for the Re-reading task, OpenHands + Claude Sonnet 4 unnecessarily modified an additional metadata field in one of the `.yml` files, which is used as part

Table 4: Detailed performance on **REXBENCH**, evaluated across three hint levels. Results are averaged across three runs.

| Agent | Model | Hints Level | File Recall | Execution Success | Final Success |
|---|---|---|---|---|---|
| aider | Claude 4 Sonnet | No hints | 0.86 | 0.50 | 0.19 |
| | | Hints | 0.84 | 0.47 | 0.31 |
| | | Detailed Hints | 0.86 | 0.39 | 0.25 |
| | Claude 3.7 Sonnet | No hints | 0.87 | 0.39 | 0.14 |
| | | Hints | 0.86 | 0.31 | 0.17 |
| | | Detailed Hints | 0.86 | 0.33 | 0.08 |
| | OpenAI GPT-5 | No hints | 0.81 | 0.36 | 0.06 |
| | | Hints | 0.84 | 0.42 | 0.06 |
| | | Detailed Hints | 0.84 | 0.25 | 0.14 |
| | OpenAI o1 | No hints | 0.84 | 0.22 | 0.00 |
| | | Hints | 0.78 | 0.31 | 0.00 |
| | | Detailed Hints | 0.80 | 0.39 | 0.03 |
| | OpenAI o4-mini | No hints | 0.43 | 0.19 | 0.03 |
| | | Hints | 0.43 | 0.25 | 0.19 |
| | | Detailed Hints | 0.43 | 0.31 | 0.14 |
| | DeepSeek-R1 | No hints | 0.18 | 0.00 | 0.00 |
| | | Hints | 0.13 | 0.00 | 0.00 |
| | | Detailed Hints | 0.13 | 0.00 | 0.00 |
| Claude Code | Claude 3.7 Sonnet | No hints | 0.76 | 0.33 | 0.25 |
| | | Hints | 0.84 | 0.50 | 0.28 |
| | | Detailed Hints | 0.88 | 0.42 | 0.25 |
| OpenHands | Claude 4 Sonnet | No hints | 0.79 | 0.67 | 0.31 |
| | | Hints | 0.85 | 0.67 | 0.44 |
| | | Detailed Hints | 0.90 | 0.78 | 0.47 |
| | Claude 3.7 Sonnet | No hints | 0.76 | 0.42 | 0.25 |
| | | Hints | 0.87 | 0.53 | 0.39 |
| | | Detailed Hints | 0.92 | 0.53 | 0.36 |
| | OpenAI GPT-5 | No hints | 0.79 | 0.72 | 0.31 |
| | | Hints | 0.88 | 0.61 | 0.39 |
| | | Detailed Hints | 0.87 | 0.69 | 0.42 |
| | OpenAI o1 | No hints | 0.64 | 0.31 | 0.00 |
| | | Hints | 0.67 | 0.33 | 0.08 |
| | | Detailed Hints | 0.78 | 0.39 | 0.03 |
| | OpenAI o4-mini | No hints | 0.68 | 0.39 | 0.08 |
| | | Hints | 0.77 | 0.36 | 0.19 |
| | | Detailed Hints | 0.74 | 0.47 | 0.14 |
| | DeepSeek-R1 | No hints | 0.74 | 0.14 | 0.00 |
| | | Hints | 0.76 | 0.14 | 0.00 |
| | | Detailed Hints | 0.71 | 0.22 | 0.08 |

of the input in one of the experimental settings. Similarly, in the VariErr-NLI task OpenHands + GPT-5 unnecessarily modified an output file path required for obtaining the final scores, resulting in the evaluation scripts being unable to access the results of the agent's implementation. Given that scientific work relies on rigor and reproducibility, deviations like these from the specified instructions are problematic. This highlights the need to design agents which conform exactly to the requirements given, without introducing additional unrequested changes.

## H  LICENSE INFORMATION

The codebase portion of **REXBENCH** is constructed from public repositories—details of the licenses for each task are provided in Table 22. When the codebase did not contain any license information, we reached out to the authors for more information and used their suggestion (one response still pending at the time of writing, but we make an educated guess that the repository will be associated with a permissive license given that the paper was written by authors with primarily academic affiliations,

Table 5: Token usage statistics across agents and models.

| Agent | Model | Hints Level | Total Turns (Avg.) | Prompt Tokens (Avg.) | Output Tokens (Avg.) |
|---|---|---|---|---|---|
| aider | Claude 4 Sonnet | No hints | 2.00 | 3121.10 | 3694.40 |
| | | Hints | 2.00 | 3055.70 | 3794.00 |
| | | Detailed Hints | 2.00 | 3303.80 | 4254.70 |
| | Claude 3.7 Sonnet | No hints | 2.00 | 3053.60 | 5204.20 |
| | | Hints | 2.00 | 3029.10 | 4222.20 |
| | | Detailed Hints | 2.00 | 3529.20 | 3996.20 |
| | OpenAI GPT-5 | No hints | 2.00 | 3280.60 | 8995.60 |
| | | Hints | 2.00 | 3221.10 | 8544.10 |
| | | Detailed Hints | 2.00 | 3466.40 | 9697.80 |
| | OpenAI o1 | No hints | 2.00 | 2964.60 | 5302.20 |
| | | Hints | 2.00 | 3061.50 | 6052.80 |
| | | Detailed Hints | 2.00 | 3447.60 | 6011.10 |
| | OpenAI o4-mini | No hints | 2.00 | 2910.60 | 4286.10 |
| | | Hints | 2.00 | 3002.80 | 2875.30 |
| | | Detailed Hints | 2.00 | 4457.30 | 3388.80 |
| | DeepSeek-R1 | No hints | 2.00 | 2963.80 | 3557.10 |
| | | Hints | 2.00 | 3045.80 | 3751.50 |
| | | Detailed Hints | 2.00 | 3446.20 | 3378.70 |
| Claude Code | Claude 3.7 Sonnet | No hints | 34.64 | – | – |
| | | Hints | 29.61 | – | – |
| | | Detailed Hints | 25.00 | – | – |
| OpenHands | Claude 4 Sonnet | No hints | 94.53 | 1,455,087.28 | 10,080.81 |
| | | Hints | 88.97 | 1,519,934.47 | 9370.17 |
| | | Detailed Hints | 97.58 | 1,866,521.03 | 9738.08 |
| | Claude 3.7 Sonnet | No hints | 50.94 | 542,311.69 | 7492.75 |
| | | Hints | 47.92 | 540,580.78 | 7116.36 |
| | | Detailed Hints | 43.72 | 458,430.67 | 6882.17 |
| | OpenAI GPT-5 | No hints | 67.25 | 973,985.28 | 34,009.06 |
| | | Hints | 72.72 | 1,011,380.39 | 40,850.58 |
| | | Detailed Hints | 70.44 | 882,550.53 | 37,924.22 |
| | OpenAI o1 | No hints | 16.58 | 82,144.81 | 9592.97 |
| | | Hints | 23.61 | 137,508.86 | 13,816.28 |
| | | Detailed Hints | 27.94 | 183,731.94 | 18,861.33 |
| | OpenAI o4-mini | No hints | 53.47 | 522,734.39 | 25,140.58 |
| | | Hints | 54.64 | 511,121.25 | 25,899.81 |
| | | Detailed Hints | 57.72 | 565,430.33 | 26,719.36 |
| | DeepSeek-R1 | No hints | 34.00 | 194,548.08 | 18,099.22 |
| | | Hints | 34.78 | 202,214.92 | 19,673.08 |
| | | Detailed Hints | 36.19 | 293,836.28 | 20,307.64 |

and from the fact that the public availability of their codebase is mentioned in the paper). We release our data and code under a dual license (MIT and Apache 2.0), given the mixed license of the repositories included in the full benchmark suite.

Table 6: Cost and duration statistics across agents and models (main experiment).

| Agent | Model | Avg. Cost ($) | Avg. Duration | Total Cost ($) | Total Duration |
|---|---|---|---|---|---|
| aider | Claude 4 Sonnet | 0.19 | 1m 4s | 6.99 | 38m 27s |
| | Claude 3.7 Sonnet | 0.41 | 1m 44s | 14.77 | 1h 2m 41s |
| | OpenAI GPT-5 | 0.10 | 2m 58s | 3.61 | 1h 46m 53s |
| | OpenAI o1 | 0.62 | 3m 7s | 22.37 | 1h 51m 59s |
| | OpenAI o4-mini | 0.03 | 42s | 1.02 | 24m 56s |
| | DeepSeek-R1 | 0.04 | 4m 38s | 1.46 | 2h 42m 10s |
| Claude Code | Claude 3.7 Sonnet | 0.60 | 2m 45s | 21.94 | 1h 38m 45s |
| OpenHands | Claude 4 Sonnet | 4.52 | 6m 17s | 162.59 | 3h 46m 16s |
| | Claude 3.7 Sonnet | 0.40 | 2m 43s | 14.22 | 1h 38m 6s |
| | OpenAI GPT-5 | 1.14 | 12m 52s | 41.20 | 7h 43m 14s |
| | OpenAI o1 | 1.30 | 2m 25s | 46.93 | 1h 27m 4s |
| | OpenAI o4-mini | 0.61 | 5m 2s | 22.09 | 3h 1m 45s |
| | DeepSeek-v3 | 0.03 | 4m 41s | 1.01 | 2h 49m 5s |

Table 7: Breakdown of error counts for Aider and Claude Code.

| Error Type | aider | | | | | | Claude Code |
|---|---|---|---|---|---|---|---|
| | Claude 4 Sonnet | Claude 3.7 Sonnet | OpenAI GPT-5 | OpenAI o1 | OpenAI o4-mini | DeepSeek R1 | Claude 3.7 Sonnet |
| ***Python Errors*** | | | | | | | |
| AssertionError | 1 | 0 | 4 | 0 | 1 | 0 | 0 |
| AttributeError | 3 | 11 | 4 | 5 | 2 | 0 | 3 |
| FileNotFoundError | 14 | 23 | 7 | 11 | 4 | 2 | 6 |
| ImportError | 1 | 6 | 2 | 6 | 0 | 0 | 2 |
| IndentationError | 0 | 0 | 0 | 0 | 0 | 0 | 0 |
| IndexError | 0 | 0 | 1 | 1 | 0 | 0 | 1 |
| IsADirectoryError | 0 | 0 | 0 | 0 | 0 | 0 | 0 |
| KeyError | 0 | 5 | 1 | 2 | 6 | 1 | 4 |
| LookupError | 0 | 0 | 1 | 0 | 0 | 0 | 0 |
| ModuleNotFoundError | 2 | 0 | 5 | 3 | 0 | 0 | 0 |
| NameError | 1 | 1 | 1 | 1 | 0 | 1 | 0 |
| NotImplementedError | 0 | 0 | 0 | 0 | 0 | 0 | 0 |
| OSError | 0 | 0 | 0 | 0 | 0 | 0 | 0 |
| RuntimeError | 0 | 0 | 1 | 1 | 0 | 0 | 0 |
| SyntaxError | 1 | 0 | 0 | 5 | 4 | 0 | 2 |
| TypeError | 1 | 3 | 4 | 2 | 6 | 3 | 10 |
| UnboundLocalError | 1 | 0 | 0 | 1 | 0 | 0 | 0 |
| ValueError | 28 | 10 | 30 | 8 | 6 | 4 | 24 |
| EOFError | 0 | 1 | 0 | 0 | 0 | 0 | 0 |
| ***Python Library Errors*** | | | | | | | |
| DatasetNotFoundError | 1 | 1 | 0 | 0 | 0 | 0 | 0 |
| NotFoundError | 2 | 0 | 1 | 1 | 0 | 0 | 1 |
| OutOfMemory | 0 | 0 | 0 | 0 | 0 | 0 | 0 |
| ArgumentError | 0 | 0 | 0 | 0 | 0 | 0 | 0 |
| ScannerError | 0 | 0 | 0 | 0 | 0 | 0 | 1 |
| ***Other Python Errors*** | | | | | | | |
| ConstructorError | 1 | 0 | 0 | 0 | 0 | 0 | 0 |
| JSONDecodeError | 2 | 0 | 3 | 0 | 0 | 0 | 0 |
| ***Bash Errors*** | | | | | | | |
| cannot create directory | 0 | 0 | 0 | 1 | 0 | 0 | 0 |
| empty patch | 0 | 0 | 0 | 0 | 35 | 59 | 0 |
| empty or missing | 0 | 0 | 2 | 4 | 5 | 25 | 0 |
| unable to write file | 0 | 0 | 0 | 0 | 0 | 3 | 0 |
| Permission denied | 0 | 0 | 0 | 2 | 0 | 0 | 0 |
| syntax error | 0 | 0 | 0 | 0 | 0 | 0 | 1 |
| cannot access | 0 | 0 | 0 | 0 | 0 | 0 | 0 |
| ***Execution Timeout*** | 0 | 2 | 0 | 7 | 1 | 1 | 4 |

Table 8: Breakdown of error counts for OpenHands.

| Error Type | OpenHands | | | | | |
|---|---|---|---|---|---|---|
| | Claude 4 Sonnet | Claude 3.7 Sonnet | OpenAI GPT-5 | OpenAI o1 | OpenAI o4-mini | DeepSeek R1 |
| ***Python Errors*** | | | | | | |
| AssertionError | 0 | 0 | 1 | 2 | 2 | 0 |
| AttributeError | 0 | 7 | 1 | 2 | 2 | 5 |
| FileNotFoundError | 3 | 4 | 0 | 3 | 1 | 3 |
| ImportError | 0 | 0 | 0 | 0 | 1 | 3 |
| IndentationError | 0 | 0 | 3 | 10 | 8 | 11 |
| IndexError | 3 | 1 | 0 | 1 | 0 | 1 |
| IsADirectoryError | 0 | 0 | 0 | 0 | 0 | 1 |
| KeyError | 1 | 4 | 1 | 1 | 2 | 10 |
| LookupError | 0 | 0 | 0 | 0 | 0 | 0 |
| ModuleNotFoundError | 4 | 3 | 1 | 0 | 1 | 1 |
| NameError | 0 | 0 | 2 | 5 | 2 | 4 |
| NotImplementedError | 0 | 0 | 0 | 0 | 1 | 0 |
| OSError | 0 | 0 | 0 | 0 | 0 | 1 |
| RuntimeError | 0 | 0 | 0 | 0 | 0 | 0 |
| SyntaxError | 0 | 0 | 0 | 13 | 7 | 5 |
| TypeError | 5 | 13 | 2 | 2 | 7 | 7 |
| UnboundLocalError | 0 | 0 | 0 | 0 | 0 | 0 |
| ValueError | 11 | 11 | 17 | 4 | 3 | 8 |
| EOFError | 0 | 0 | 0 | 0 | 0 | 0 |
| ***Python Library Errors*** | | | | | | |
| DatasetNotFoundError | 0 | 0 | 0 | 0 | 0 | 0 |
| NotFoundError | 0 | 0 | 0 | 1 | 0 | 0 |
| OutOfMemory | 0 | 0 | 0 | 0 | 2 | 0 |
| ArgumentError | 0 | 0 | 0 | 0 | 5 | 0 |
| ScannerError | 0 | 0 | 0 | 0 | 0 | 0 |
| ***Other Python Errors*** | | | | | | |
| ConstructorError | 0 | 0 | 0 | 0 | 0 | 0 |
| JSONDecodeError | 0 | 0 | 0 | 0 | 0 | 0 |
| ***Bash Errors*** | | | | | | |
| cannot create directory | 0 | 0 | 0 | 1 | 0 | 0 |
| empty patch | 0 | 0 | 0 | 9 | 10 | 9 |
| empty or missing | 0 | 1 | 0 | 3 | 7 | 4 |
| unable to write file | 0 | 0 | 0 | 0 | 0 | 0 |
| Permission denied | 0 | 0 | 0 | 1 | 0 | 1 |
| syntax error | 3 | 0 | 0 | 0 | 0 | 0 |
| cannot access | 0 | 1 | 0 | 0 | 0 | 0 |
| ***Execution Timeout*** | 0 | 0 | 2 | 4 | 2 | 1 |

Table 9: Detailed performance on aider + Claude 4 Sonnet.

| Agent | Model | Hint Level | Task | File Recall | Execution Success | Final Success |
|-------|-------|-----------|------|-------------|-------------------|---------------|
| aider | Claude 4 Sonnet | No Hints | CheckEval | 1.00 | 1.00 | 0.00 |
| | | | COGS | 0.50 | 0.00 | 0.00 |
| | | | Entity Tracking | 1.00 | 0.67 | 0.33 |
| | | | Explain then Translate | 1.00 | 1.00 | 0.00 |
| | | | Instruction Tuning | 0.50 | 0.00 | 0.00 |
| | | | Mission Impossible | 1.00 | 1.00 | 0.00 |
| | | | Othello | 1.00 | 1.00 | 1.00 |
| | | | Reasoning or Reciting | 1.00 | 0.00 | 0.00 |
| | | | Re-reading | 0.52 | 0.33 | 0.00 |
| | | | Tree of Thoughts | 1.00 | 0.00 | 0.00 |
| | | | VariErr-NLI | 1.00 | 0.00 | 0.00 |
| | | | WinoDict | 0.75 | 1.00 | 1.00 |
| | | Hints | CheckEval | 1.00 | 1.00 | 0.00 |
| | | | COGS | 0.50 | 0.00 | 0.00 |
| | | | Entity Tracking | 1.00 | 1.00 | 1.00 |
| | | | Explain then Translate | 0.80 | 0.00 | 0.00 |
| | | | Instruction Tuning | 0.50 | 0.00 | 0.00 |
| | | | Mission Impossible | 1.00 | 1.00 | 0.67 |
| | | | Othello | 1.00 | 1.00 | 1.00 |
| | | | Reasoning or Reciting | 1.00 | 0.33 | 0.00 |
| | | | Re-reading | 0.60 | 0.33 | 0.00 |
| | | | Tree of Thoughts | 1.00 | 0.00 | 0.00 |
| | | | VariErr-NLI | 0.90 | 0.00 | 0.00 |
| | | | WinoDict | 0.75 | 1.00 | 1.00 |
| | | Detailed Hints | CheckEval | 1.00 | 0.33 | 0.00 |
| | | | COGS | 1.00 | 0.00 | 0.00 |
| | | | Entity Tracking | 1.00 | 0.33 | 0.00 |
| | | | Explain then Translate | 1.00 | 1.00 | 1.00 |
| | | | Instruction Tuning | 0.40 | 0.00 | 0.00 |
| | | | Mission Impossible | 1.00 | 0.67 | 0.33 |
| | | | Othello | 1.00 | 1.00 | 0.67 |
| | | | Reasoning or Reciting | 1.00 | 0.00 | 0.00 |
| | | | Re-reading | 0.60 | 0.33 | 0.00 |
| | | | Tree of Thoughts | 0.60 | 0.00 | 0.00 |
| | | | VariErr-NLI | 1.00 | 0.00 | 0.00 |
| | | | WinoDict | 0.75 | 1.00 | 1.00 |

Table 10: Detailed performance on aider + Claude 3.7 Sonnet.

| Agent | Model | Hint Level | Task | File Recall | Execution Success | Final Success |
|-------|-------|-----------|------|-------------|-------------------|---------------|
| aider | Claude 3.7 Sonnet | No Hints | CheckEval | 1.00 | 0.00 | 0.00 |
| | | | COGS | 0.50 | 0.00 | 0.00 |
| | | | Entity Tracking | 1.00 | 0.33 | 0.33 |
| | | | Explain then Translate | 1.00 | 1.00 | 0.00 |
| | | | Instruction Tuning | 0.50 | 0.67 | 0.00 |
| | | | Mission Impossible | 1.00 | 1.00 | 0.33 |
| | | | Othello | 1.00 | 1.00 | 1.00 |
| | | | Reasoning or Reciting | 0.60 | 0.00 | 0.00 |
| | | | Re-reading | 1.00 | 0.33 | 0.00 |
| | | | Tree of Thoughts | 1.00 | 0.00 | 0.00 |
| | | | VariErr-NLI | 1.00 | 0.00 | 0.00 |
| | | | WinoDict | 0.75 | 0.33 | 0.00 |
| | | Hints | CheckEval | 1.00 | 0.00 | 0.00 |
| | | | COGS | 0.50 | 0.00 | 0.00 |
| | | | Entity Tracking | 1.00 | 0.67 | 0.33 |
| | | | Explain then Translate | 1.00 | 1.00 | 0.00 |
| | | | Instruction Tuning | 0.50 | 0.00 | 0.00 |
| | | | Mission Impossible | 1.00 | 1.00 | 0.67 |
| | | | Othello | 1.00 | 1.00 | 1.00 |
| | | | Reasoning or Reciting | 0.40 | 0.00 | 0.00 |
| | | | Re-reading | 1.00 | 0.00 | 0.00 |
| | | | Tree of Thoughts | 1.00 | 0.00 | 0.00 |
| | | | VariErr-NLI | 1.00 | 0.00 | 0.00 |
| | | | WinoDict | 0.75 | 0.00 | 0.00 |
| | | Detailed Hints | CheckEval | 1.00 | 0.67 | 0.00 |
| | | | COGS | 0.50 | 0.00 | 0.00 |
| | | | Entity Tracking | 1.00 | 0.00 | 0.00 |
| | | | Explain then Translate | 1.00 | 1.00 | 1.00 |
| | | | Instruction Tuning | 0.50 | 0.00 | 0.00 |
| | | | Mission Impossible | 1.00 | 1.00 | 0.00 |
| | | | Othello | 1.00 | 1.00 | 0.00 |
| | | | Reasoning or Reciting | 0.40 | 0.00 | 0.00 |
| | | | Re-reading | 1.00 | 0.00 | 0.00 |
| | | | Tree of Thoughts | 1.00 | 0.00 | 0.00 |
| | | | VariErr-NLI | 1.00 | 0.00 | 0.00 |
| | | | WinoDict | 0.75 | 0.33 | 0.00 |

Table 11: Detailed performance on aider + GPT-5.

| Agent | Model | Hint Level | Task | File Recall | Execution Success | Final Success |
|-------|-------|------------|------|-------------|-------------------|---------------|
| aider | GPT-5 | No Hints | CheckEval | 0.80 | 0.33 | 0.00 |
| | | | COGS | 1.00 | 0.33 | 0.00 |
| | | | Entity Tracking | 1.00 | 0.33 | 0.00 |
| | | | Explain then Translate | 1.00 | 1.00 | 0.00 |
| | | | Instruction Tuning | 1.00 | 0.00 | 0.00 |
| | | | Mission Impossible | 1.00 | 0.33 | 0.00 |
| | | | Othello | 1.00 | 0.67 | 0.00 |
| | | | Reasoning or Reciting | 0.20 | 0.33 | 0.00 |
| | | | Re-reading | 0.40 | 0.00 | 0.00 |
| | | | Tree of Thoughts | 0.90 | 0.00 | 0.00 |
| | | | VariErr-NLI | 1.00 | 0.00 | 0.00 |
| | | | WinoDict | 0.60 | 1.00 | 0.67 |
| | | Hints | CheckEval | 1.00 | 1.00 | 0.00 |
| | | | COGS | 1.00 | 0.33 | 0.00 |
| | | | Entity Tracking | 1.00 | 0.67 | 0.00 |
| | | | Explain then Translate | 1.00 | 1.00 | 0.00 |
| | | | Instruction Tuning | 1.00 | 0.00 | 0.00 |
| | | | Mission Impossible | 1.00 | 0.33 | 0.00 |
| | | | Othello | 1.00 | 0.67 | 0.00 |
| | | | Reasoning or Reciting | 0.20 | 0.00 | 0.00 |
| | | | Re-reading | 0.56 | 0.33 | 0.00 |
| | | | Tree of Thoughts | 0.70 | 0.00 | 0.00 |
| | | | VariErr-NLI | 1.00 | 0.00 | 0.00 |
| | | | WinoDict | 0.75 | 0.67 | 0.67 |
| | | Detailed Hints | CheckEval | 1.00 | 0.00 | 0.00 |
| | | | COGS | 1.00 | 1.00 | 0.00 |
| | | | Entity Tracking | 0.80 | 0.00 | 0.00 |
| | | | Explain then Translate | 1.00 | 1.00 | 0.67 |
| | | | Instruction Tuning | 1.00 | 0.00 | 0.00 |
| | | | Mission Impossible | 1.00 | 0.33 | 0.00 |
| | | | Othello | 1.00 | 0.00 | 0.00 |
| | | | Reasoning or Reciting | 0.20 | 0.00 | 0.00 |
| | | | Re-reading | 0.48 | 0.00 | 0.00 |
| | | | Tree of Thoughts | 1.00 | 0.00 | 0.00 |
| | | | VariErr-NLI | 1.00 | 0.00 | 0.00 |
| | | | WinoDict | 0.75 | 1.00 | 1.00 |

Table 12: Detailed performance on aider + o1.

| Agent | Model | Hint Level | Task | File Recall | Execution Success | Final Success |
|-------|-------|-----------|------|-------------|-------------------|---------------|
| aider | o1 | No Hints | CheckEval | 0.83 | 0.33 | 0.00 |
| | | | COGS | 1.00 | 0.00 | 0.00 |
| | | | Entity Tracking | 1.00 | 1.00 | 0.00 |
| | | | Explain then Translate | 1.00 | 0.33 | 0.00 |
| | | | Instruction Tuning | 0.00 | 0.00 | 0.00 |
| | | | Mission Impossible | 1.00 | 0.33 | 0.00 |
| | | | Othello | 1.00 | 0.00 | 0.00 |
| | | | Reasoning or Reciting | 0.33 | 0.00 | 0.00 |
| | | | Re-reading | 1.00 | 0.00 | 0.00 |
| | | | Tree of Thoughts | 1.00 | 0.00 | 0.00 |
| | | | VariErr-NLI | 1.00 | 0.00 | 0.00 |
| | | | WinoDict | 0.75 | 0.67 | 0.00 |
| | | Hints | CheckEval | 0.67 | 0.67 | 0.00 |
| | | | COGS | 1.00 | 0.00 | 0.00 |
| | | | Entity Tracking | 1.00 | 1.00 | 0.00 |
| | | | Explain then Translate | 1.00 | 0.00 | 0.00 |
| | | | Instruction Tuning | 0.00 | 0.00 | 0.00 |
| | | | Mission Impossible | 1.00 | 0.67 | 0.00 |
| | | | Othello | 1.00 | 0.33 | 0.00 |
| | | | Reasoning or Reciting | 0.60 | 0.00 | 0.00 |
| | | | Re-reading | 1.00 | 0.00 | 0.00 |
| | | | Tree of Thoughts | 1.00 | 1.00 | 0.00 |
| | | | VariErr-NLI | 1.00 | 0.00 | 0.00 |
| | | | WinoDict | 0.67 | 0.67 | 0.00 |
| | | Detailed Hints | CheckEval | 0.67 | 0.00 | 0.00 |
| | | | COGS | 1.00 | 0.67 | 0.00 |
| | | | Entity Tracking | 1.00 | 1.00 | 0.00 |
| | | | Explain then Translate | 1.00 | 0.33 | 0.33 |
| | | | Instruction Tuning | 0.00 | 0.00 | 0.00 |
| | | | Mission Impossible | 1.00 | 0.67 | 0.00 |
| | | | Othello | 1.00 | 1.00 | 0.00 |
| | | | Reasoning or Reciting | 0.47 | 0.00 | 0.00 |
| | | | Re-reading | 1.00 | 0.33 | 0.00 |
| | | | Tree of Thoughts | 0.83 | 0.00 | 0.00 |
| | | | VariErr-NLI | 1.00 | 0.00 | 0.00 |
| | | | WinoDict | 0.75 | 0.67 | 0.00 |

Table 13: Detailed performance on aider + o4-mini.

| Agent | Model | Hint Level | Task | File Recall | Execution Success | Final Success |
|-------|-------|------------|------|-------------|-------------------|---------------|
| aider | o4-mini | No Hints | CheckEval | 0.67 | 0.00 | 0.00 |
| | | | COGS | 0.00 | 0.00 | 0.00 |
| | | | Entity Tracking | 0.67 | 0.67 | 0.00 |
| | | | Explain then Translate | 0.67 | 0.33 | 0.00 |
| | | | Instruction Tuning | 0.00 | 0.00 | 0.00 |
| | | | Mission Impossible | 0.83 | 0.33 | 0.33 |
| | | | Othello | 1.00 | 0.33 | 0.00 |
| | | | Reasoning or Reciting | 0.27 | 0.00 | 0.00 |
| | | | Re-reading | 0.00 | 0.00 | 0.00 |
| | | | Tree of Thoughts | 0.83 | 0.00 | 0.00 |
| | | | VariErr-NLI | 0.33 | 0.00 | 0.00 |
| | | | WinoDict | 0.25 | 0.33 | 0.00 |
| | | Hints | CheckEval | 0.00 | 0.00 | 0.00 |
| | | | COGS | 0.00 | 0.00 | 0.00 |
| | | | Entity Tracking | 1.00 | 1.00 | 1.00 |
| | | | Explain then Translate | 1.00 | 0.00 | 0.00 |
| | | | Instruction Tuning | 0.00 | 0.00 | 0.00 |
| | | | Mission Impossible | 1.00 | 0.67 | 0.33 |
| | | | Othello | 1.00 | 1.00 | 1.00 |
| | | | Reasoning or Reciting | 0.33 | 0.00 | 0.00 |
| | | | Re-reading | 0.00 | 0.00 | 0.00 |
| | | | Tree of Thoughts | 0.17 | 0.00 | 0.00 |
| | | | VariErr-NLI | 0.33 | 0.00 | 0.00 |
| | | | WinoDict | 0.58 | 0.33 | 0.00 |
| | | Detailed Hints | CheckEval | 0.33 | 0.00 | 0.00 |
| | | | COGS | 0.33 | 0.67 | 0.00 |
| | | | Entity Tracking | 1.00 | 1.00 | 0.67 |
| | | | Explain then Translate | 0.67 | 0.67 | 0.33 |
| | | | Instruction Tuning | 0.17 | 0.33 | 0.00 |
| | | | Mission Impossible | 0.67 | 0.33 | 0.00 |
| | | | Othello | 0.67 | 0.67 | 0.67 |
| | | | Reasoning or Reciting | 0.47 | 0.00 | 0.00 |
| | | | Re-reading | 0.00 | 0.00 | 0.00 |
| | | | Tree of Thoughts | 0.67 | 0.00 | 0.00 |
| | | | VariErr-NLI | 0.67 | 0.00 | 0.00 |
| | | | WinoDict | 0.42 | 0.00 | 0.00 |

Table 14: Detailed performance on aider + Deepseek-R1.

| Agent | Model | Hint Level | Task | File Recall | Execution Success | Final Success |
|-------|-------|------------|------|-------------|-------------------|---------------|
| aider | Deepseek-R1 | No Hints | CheckEval | 0.00 | 0.00 | 0.00 |
| | | | COGS | 0.00 | 0.00 | 0.00 |
| | | | Entity Tracking | 0.00 | 0.00 | 0.00 |
| | | | Explain then Translate | 0.00 | 0.00 | 0.00 |
| | | | Instruction Tuning | 0.33 | 0.00 | 0.00 |
| | | | Mission Impossible | 0.67 | 0.00 | 0.00 |
| | | | Othello | 0.00 | 0.00 | 0.00 |
| | | | Reasoning or Reciting | 0.00 | 0.00 | 0.00 |
| | | | Re-reading | 0.00 | 0.00 | 0.00 |
| | | | Tree of Thoughts | 0.00 | 0.00 | 0.00 |
| | | | VariErr-NLI | 0.33 | 0.00 | 0.00 |
| | | | WinoDict | 0.00 | 0.00 | 0.00 |
| | | Hints | CheckEval | 0.33 | 0.00 | 0.00 |
| | | | COGS | 0.00 | 0.00 | 0.00 |
| | | | Entity Tracking | 0.00 | 0.00 | 0.00 |
| | | | Explain then Translate | 0.00 | 0.00 | 0.00 |
| | | | Instruction Tuning | 0.00 | 0.00 | 0.00 |
| | | | Mission Impossible | 0.00 | 0.00 | 0.00 |
| | | | Othello | 0.00 | 0.00 | 0.00 |
| | | | Reasoning or Reciting | 0.00 | 0.00 | 0.00 |
| | | | Re-reading | 0.00 | 0.00 | 0.00 |
| | | | Tree of Thoughts | 0.00 | 0.00 | 0.00 |
| | | | VariErr-NLI | 0.33 | 0.00 | 0.00 |
| | | | WinoDict | 0.00 | 0.00 | 0.00 |
| | | Detailed Hints | CheckEval | 0.00 | 0.00 | 0.00 |
| | | | COGS | 0.00 | 0.00 | 0.00 |
| | | | Entity Tracking | 0.33 | 0.00 | 0.00 |
| | | | Explain then Translate | 0.00 | 0.00 | 0.00 |
| | | | Instruction Tuning | 0.00 | 0.00 | 0.00 |
| | | | Mission Impossible | 0.00 | 0.00 | 0.00 |
| | | | Othello | 0.00 | 0.00 | 0.00 |
| | | | Reasoning or Reciting | 0.13 | 0.00 | 0.00 |
| | | | Re-reading | 0.00 | 0.00 | 0.00 |
| | | | Tree of Thoughts | 0.00 | 0.00 | 0.00 |
| | | | VariErr-NLI | 0.67 | 0.00 | 0.00 |
| | | | WinoDict | 0.00 | 0.00 | 0.00 |

Table 15: Detailed performance on Claude Code + Claude 3.7 Sonnet.

| Agent | Model | Hint Level | Task | File Recall | Execution Success | Final Success |
|---|---|---|---|---|---|---|
| Claude-Code | Claude 3.7 Sonnet | No Hints | CheckEval | 0.50 | 0.00 | 0.00 |
| | | | COGS | 0.50 | 1.00 | 1.00 |
| | | | Entity Tracking | 1.00 | 0.00 | 0.00 |
| | | | Explain then Translate | 1.00 | 1.00 | 0.67 |
| | | | Instruction Tuning | 0.83 | 0.00 | 0.00 |
| | | | Mission Impossible | 0.67 | 0.33 | 0.00 |
| | | | Othello | 1.00 | 1.00 | 1.00 |
| | | | Reasoning or Reciting | 0.40 | 0.33 | 0.00 |
| | | | Re-reading | 1.00 | 0.00 | 0.00 |
| | | | Tree of Thoughts | 1.00 | 0.00 | 0.00 |
| | | | VariErr-NLI | 1.00 | 0.00 | 0.00 |
| | | | WinoDict | 0.25 | 0.33 | 0.33 |
| | | Hints | CheckEval | 0.50 | 0.00 | 0.00 |
| | | | COGS | 0.50 | 1.00 | 1.00 |
| | | | Entity Tracking | 1.00 | 0.00 | 0.00 |
| | | | Explain then Translate | 1.00 | 1.00 | 0.33 |
| | | | Instruction Tuning | 0.83 | 0.33 | 0.33 |
| | | | Mission Impossible | 1.00 | 0.67 | 0.00 |
| | | | Othello | 1.00 | 1.00 | 1.00 |
| | | | Reasoning or Reciting | 0.40 | 0.00 | 0.00 |
| | | | Re-reading | 1.00 | 0.67 | 0.00 |
| | | | Tree of Thoughts | 1.00 | 1.00 | 0.67 |
| | | | VariErr-NLI | 1.00 | 0.67 | 0.00 |
| | | | WinoDict | 0.75 | 0.00 | 0.00 |
| | | Detailed Hints | CheckEval | 0.83 | 0.00 | 0.00 |
| | | | COGS | 0.50 | 1.00 | 1.00 |
| | | | Entity Tracking | 1.00 | 0.33 | 0.33 |
| | | | Explain then Translate | 1.00 | 1.00 | 1.00 |
| | | | Instruction Tuning | 1.00 | 0.33 | 0.00 |
| | | | Mission Impossible | 1.00 | 1.00 | 0.67 |
| | | | Othello | 1.00 | 0.67 | 0.00 |
| | | | Reasoning or Reciting | 0.40 | 0.33 | 0.00 |
| | | | Re-reading | 1.00 | 0.33 | 0.00 |
| | | | Tree of Thoughts | 1.00 | 0.00 | 0.00 |
| | | | VariErr-NLI | 1.00 | 0.00 | 0.00 |
| | | | WinoDict | 0.75 | 0.33 | 0.00 |

Table 16: Detailed performance on OpenHands + Claude 4 Sonnet.

| Agent | Model | Hint Level | Task | File Recall | Execution Success | Final Success |
|---|---|---|---|---|---|---|
| OpenHands | Claude 4 Sonnet | No Hints | CheckEval | 0.40 | 0.67 | 0.00 |
| | | | COGS | 0.60 | 1.00 | 1.00 |
| | | | Entity Tracking | 1.00 | 0.00 | 0.00 |
| | | | Explain then Translate | 1.00 | 1.00 | 0.67 |
| | | | Instruction Tuning | 1.00 | 0.33 | 0.33 |
| | | | Mission Impossible | 0.80 | 0.67 | 0.00 |
| | | | Othello | 1.00 | 1.00 | 1.00 |
| | | | Reasoning or Reciting | 1.00 | 0.67 | 0.00 |
| | | | Re-reading | 0.40 | 0.67 | 0.00 |
| | | | Tree of Thoughts | 1.00 | 1.00 | 0.33 |
| | | | VariErr-NLI | 1.00 | 0.00 | 0.00 |
| | | | WinoDict | 0.25 | 1.00 | 0.33 |
| | | Hints | CheckEval | 0.60 | 0.00 | 0.00 |
| | | | COGS | 0.50 | 0.33 | 0.33 |
| | | | Entity Tracking | 1.00 | 1.00 | 1.00 |
| | | | Explain then Translate | 1.00 | 1.00 | 1.00 |
| | | | Instruction Tuning | 1.00 | 0.67 | 0.67 |
| | | | Mission Impossible | 1.00 | 1.00 | 0.67 |
| | | | Othello | 1.00 | 1.00 | 1.00 |
| | | | Reasoning or Reciting | 1.00 | 0.67 | 0.00 |
| | | | Re-reading | 0.40 | 0.67 | 0.00 |
| | | | Tree of Thoughts | 1.00 | 1.00 | 0.33 |
| | | | VariErr-NLI | 1.00 | 0.00 | 0.00 |
| | | | WinoDict | 0.65 | 0.67 | 0.33 |
| | | Detailed Hints | CheckEval | 0.60 | 1.00 | 0.67 |
| | | | COGS | 1.00 | 0.67 | 0.67 |
| | | | Entity Tracking | 1.00 | 1.00 | 0.33 |
| | | | Explain then Translate | 1.00 | 1.00 | 1.00 |
| | | | Instruction Tuning | 1.00 | 0.00 | 0.33 |
| | | | Mission Impossible | 1.00 | 1.00 | 0.67 |
| | | | Othello | 1.00 | 1.00 | 1.00 |
| | | | Reasoning or Reciting | 1.00 | 1.00 | 0.00 |
| | | | Re-reading | 0.40 | 1.00 | 0.00 |
| | | | Tree of Thoughts | 1.00 | 1.00 | 0.67 |
| | | | VariErr-NLI | 1.00 | 0.00 | 0.00 |
| | | | WinoDict | 0.75 | 0.67 | 0.33 |

Table 17: Detailed performance on OpenHands + Claude 3.7 Sonnet.

| Agent | Model | Hint Level | Task | File Recall | Execution Success | Final Success |
|-------|-------|-----------|------|-------------|-------------------|---------------|
| OpenHands | Claude 3.7 Sonnet | No Hints | CheckEval | 0.50 | 0.00 | 0.00 |
| | | | COGS | 0.50 | 1.00 | 0.67 |
| | | | Entity Tracking | 1.00 | 0.00 | 0.00 |
| | | | Explain then Translate | 1.00 | 1.00 | 1.00 |
| | | | Instruction Tuning | 1.00 | 0.67 | 0.33 |
| | | | Mission Impossible | 0.50 | 0.00 | 0.00 |
| | | | Othello | 1.00 | 1.00 | 1.00 |
| | | | Reasoning or Reciting | 0.40 | 0.00 | 0.00 |
| | | | Re-reading | 1.00 | 0.33 | 0.00 |
| | | | Tree of Thoughts | 1.00 | 0.33 | 0.00 |
| | | | VariErr-NLI | 1.00 | 0.33 | 0.00 |
| | | | WinoDict | 0.25 | 0.33 | 0.00 |
| | | Hints | CheckEval | 0.67 | 0.67 | 0.33 |
| | | | COGS | 1.00 | 1.00 | 1.00 |
| | | | Entity Tracking | 1.00 | 1.00 | 1.00 |
| | | | Explain then Translate | 1.00 | 1.00 | 0.67 |
| | | | Instruction Tuning | 0.50 | 0.00 | 0.33 |
| | | | Mission Impossible | 1.00 | 0.67 | 0.00 |
| | | | Othello | 1.00 | 1.00 | 1.00 |
| | | | Reasoning or Reciting | 0.40 | 0.00 | 0.00 |
| | | | Re-reading | 1.00 | 0.00 | 0.00 |
| | | | Tree of Thoughts | 1.00 | 0.67 | 0.33 |
| | | | VariErr-NLI | 1.00 | 0.00 | 0.00 |
| | | | WinoDict | 0.75 | 0.00 | 0.00 |
| | | Detailed Hints | CheckEval | 1.00 | 0.33 | 0.00 |
| | | | COGS | 1.00 | 1.00 | 1.00 |
| | | | Entity Tracking | 1.00 | 1.00 | 1.00 |
| | | | Explain then Translate | 1.00 | 1.00 | 1.00 |
| | | | Instruction Tuning | 1.00 | 0.67 | 0.00 |
| | | | Mission Impossible | 1.00 | 1.00 | 1.00 |
| | | | Othello | 1.00 | 1.00 | 0.00 |
| | | | Reasoning or Reciting | 0.40 | 0.00 | 0.00 |
| | | | Re-reading | 1.00 | 0.00 | 0.00 |
| | | | Tree of Thoughts | 0.83 | 0.33 | 0.33 |
| | | | VariErr-NLI | 1.00 | 0.00 | 0.00 |
| | | | WinoDict | 0.75 | 0.00 | 0.00 |

Table 18: Detailed performance on OpenHands + GPT-5.

| Agent | Model | Hint Level | Task | File Recall | Execution Success | Final Success |
|---|---|---|---|---|---|---|
| OpenHands | GPT-5 | No Hints | CheckEval | 0.50 | 1.00 | 0.00 |
| | | | COGS | 0.50 | 0.67 | 0.67 |
| | | | Entity Tracking | 1.00 | 0.00 | 0.00 |
| | | | Explain then Translate | 1.00 | 1.00 | 0.67 |
| | | | Instruction Tuning | 1.00 | 0.00 | 0.00 |
| | | | Mission Impossible | 0.80 | 0.67 | 0.00 |
| | | | Othello | 1.00 | 1.00 | 1.00 |
| | | | Reasoning or Reciting | 1.00 | 1.00 | 0.00 |
| | | | Re-reading | 0.40 | 1.00 | 0.00 |
| | | | Tree of Thoughts | 1.00 | 1.00 | 0.33 |
| | | | VariErr-NLI | 1.00 | 0.33 | 0.00 |
| | | | WinoDict | 0.30 | 1.00 | 1.00 |
| | | Hints | CheckEval | 1.00 | 0.33 | 0.00 |
| | | | COGS | 0.50 | 1.00 | 1.00 |
| | | | Entity Tracking | 1.00 | 0.67 | 0.67 |
| | | | Explain then Translate | 1.00 | 1.00 | 1.00 |
| | | | Instruction Tuning | 0.90 | 0.00 | 0.00 |
| | | | Mission Impossible | 1.00 | 0.00 | 0.00 |
| | | | Othello | 1.00 | 0.67 | 0.33 |
| | | | Reasoning or Reciting | 1.00 | 0.67 | 0.00 |
| | | | Re-reading | 0.40 | 1.00 | 0.00 |
| | | | Tree of Thoughts | 1.00 | 1.00 | 0.67 |
| | | | VariErr-NLI | 1.00 | 0.33 | 0.00 |
| | | | WinoDict | 0.75 | 0.67 | 0.67 |
| | | Detailed Hints | CheckEval | 0.90 | 0.33 | 0.00 |
| | | | COGS | 0.50 | 1.00 | 1.00 |
| | | | Entity Tracking | 1.00 | 0.33 | 0.33 |
| | | | Explain then Translate | 1.00 | 1.00 | 1.00 |
| | | | Instruction Tuning | 0.90 | 0.00 | 0.00 |
| | | | Mission Impossible | 1.00 | 0.67 | 0.33 |
| | | | Othello | 1.00 | 1.00 | 1.00 |
| | | | Reasoning or Reciting | 1.00 | 1.00 | 0.00 |
| | | | Re-reading | 0.40 | 1.00 | 0.00 |
| | | | Tree of Thoughts | 1.00 | 0.67 | 0.67 |
| | | | VariErr-NLI | 1.00 | 0.33 | 0.00 |
| | | | WinoDict | 0.75 | 1.00 | 0.67 |

Table 19: Detailed performance on OpenHands + o1.

| Agent | Model | Hint Level | Task | File Recall | Execution Success | Final Success |
|---|---|---|---|---|---|---|
| OpenHands | o1 | No Hints | CheckEval | 0.50 | 1.00 | 0.00 |
| | | | COGS | 0.33 | 0.00 | 0.00 |
| | | | Entity Tracking | 1.00 | 0.33 | 0.00 |
| | | | Explain then Translate | 1.00 | 1.00 | 0.00 |
| | | | Instruction Tuning | 0.17 | 0.00 | 0.00 |
| | | | Mission Impossible | 0.50 | 0.00 | 0.00 |
| | | | Othello | 1.00 | 0.00 | 0.00 |
| | | | Reasoning or Reciting | 0.20 | 0.00 | 0.00 |
| | | | Re-reading | 1.00 | 0.00 | 0.00 |
| | | | Tree of Thoughts | 0.83 | 0.67 | 0.00 |
| | | | VariErr-NLI | 0.67 | 0.00 | 0.00 |
| | | | WinoDict | 0.25 | 0.33 | 0.00 |
| | | Hints | CheckEval | 0.50 | 1.00 | 0.00 |
| | | | COGS | 0.50 | 0.00 | 0.00 |
| | | | Entity Tracking | 1.00 | 1.00 | 1.00 |
| | | | Explain then Translate | 1.00 | 0.33 | 0.00 |
| | | | Instruction Tuning | 0.00 | 0.00 | 0.00 |
| | | | Mission Impossible | 0.50 | 0.67 | 0.00 |
| | | | Othello | 1.00 | 0.00 | 0.00 |
| | | | Reasoning or Reciting | 0.20 | 0.00 | 0.00 |
| | | | Re-reading | 1.00 | 0.00 | 0.00 |
| | | | Tree of Thoughts | 0.83 | 0.67 | 0.00 |
| | | | VariErr-NLI | 0.83 | 0.00 | 0.00 |
| | | | WinoDict | 0.17 | 0.33 | 0.00 |
| | | Detailed Hints | CheckEval | 0.83 | 0.00 | 0.00 |
| | | | COGS | 0.83 | 1.00 | 0.00 |
| | | | Entity Tracking | 0.67 | 0.67 | 0.00 |
| | | | Explain then Translate | 1.00 | 0.67 | 0.33 |
| | | | Instruction Tuning | 1.00 | 0.33 | 0.00 |
| | | | Mission Impossible | 0.67 | 0.00 | 0.00 |
| | | | Othello | 1.00 | 0.33 | 0.00 |
| | | | Reasoning or Reciting | 0.40 | 0.00 | 0.00 |
| | | | Re-reading | 1.00 | 0.00 | 0.00 |
| | | | Tree of Thoughts | 0.83 | 1.00 | 0.00 |
| | | | VariErr-NLI | 0.83 | 0.00 | 0.00 |
| | | | WinoDict | 0.25 | 0.33 | 0.00 |

Table 20: Detailed performance on OpenHands + o4-mini.

| Agent | Model | Hint Level | Task | File Recall | Execution Success | Final Success |
|---|---|---|---|---|---|---|
| OpenHands | o4-mini | No Hints | CheckEval | 0.50 | 0.33 | 0.00 |
| | | | COGS | 0.50 | 0.33 | 0.33 |
| | | | Entity Tracking | 1.00 | 1.00 | 0.33 |
| | | | Explain then Translate | 1.00 | 0.67 | 0.00 |
| | | | Instruction Tuning | 0.67 | 0.00 | 0.00 |
| | | | Mission Impossible | 0.17 | 0.00 | 0.00 |
| | | | Othello | 0.83 | 0.33 | 0.33 |
| | | | Reasoning or Reciting | 0.40 | 0.33 | 0.00 |
| | | | Re-reading | 1.00 | 0.67 | 0.00 |
| | | | Tree of Thoughts | 1.00 | 0.67 | 0.00 |
| | | | VariErr-NLI | 0.67 | 0.00 | 0.00 |
| | | | WinoDict | 0.25 | 0.33 | 0.33 |
| | | Hints | CheckEval | 0.33 | 0.00 | 0.00 |
| | | | COGS | 0.50 | 0.67 | 0.67 |
| | | | Entity Tracking | 1.00 | 1.00 | 0.67 |
| | | | Explain then Translate | 1.00 | 0.33 | 0.00 |
| | | | Instruction Tuning | 0.67 | 0.00 | 0.00 |
| | | | Mission Impossible | 0.67 | 0.67 | 0.00 |
| | | | Othello | 1.00 | 0.67 | 0.67 |
| | | | Reasoning or Reciting | 0.27 | 0.00 | 0.00 |
| | | | Re-reading | 1.00 | 1.00 | 0.33 |
| | | | Tree of Thoughts | 1.00 | 0.00 | 0.00 |
| | | | VariErr-NLI | 1.00 | 0.00 | 0.00 |
| | | | WinoDict | 0.67 | 0.00 | 0.00 |
| | | Detailed Hints | CheckEval | 0.50 | 0.33 | 0.00 |
| | | | COGS | 0.50 | 0.67 | 0.67 |
| | | | Entity Tracking | 0.67 | 0.67 | 0.00 |
| | | | Explain then Translate | 1.00 | 0.67 | 0.00 |
| | | | Instruction Tuning | 0.67 | 0.00 | 0.00 |
| | | | Mission Impossible | 1.00 | 0.67 | 0.00 |
| | | | Othello | 1.00 | 1.00 | 1.00 |
| | | | Reasoning or Reciting | 0.40 | 0.67 | 0.00 |
| | | | Re-reading | 1.00 | 1.00 | 0.00 |
| | | | Tree of Thoughts | 0.67 | 0.00 | 0.00 |
| | | | VariErr-NLI | 0.50 | 0.00 | 0.00 |
| | | | WinoDict | 0.67 | 0.00 | 0.00 |

Table 21: Detailed performance on OpenHands + Deepseek-R1.

| Agent | Model | Hint Level | Task | File Recall | Execution Success | Final Success |
|-------|-------|-----------|------|-------------|-------------------|---------------|
| OpenHands | Deepseek-R1 | No Hints | CheckEval | 0.50 | 0.00 | 0.00 |
| | | | COGS | 0.50 | 0.33 | 0.00 |
| | | | Entity Tracking | 0.67 | 0.00 | 0.00 |
| | | | Explain then Translate | 1.00 | 0.33 | 0.00 |
| | | | Instruction Tuning | 0.33 | 0.33 | 0.00 |
| | | | Mission Impossible | 0.33 | 0.00 | 0.00 |
| | | | Othello | 1.00 | 0.33 | 0.00 |
| | | | Reasoning or Reciting | 0.27 | 0.00 | 0.00 |
| | | | Re-reading | 0.67 | 0.00 | 0.00 |
| | | | Tree of Thoughts | 1.00 | 0.33 | 0.00 |
| | | | VariErr-NLI | 0.83 | 0.00 | 0.00 |
| | | | WinoDict | 0.25 | 0.00 | 0.00 |
| | | Hints | CheckEval | 0.83 | 0.33 | 0.00 |
| | | | COGS | 0.50 | 0.00 | 0.00 |
| | | | Entity Tracking | 0.67 | 0.00 | 0.00 |
| | | | Explain then Translate | 1.00 | 0.67 | 0.00 |
| | | | Instruction Tuning | 0.67 | 0.00 | 0.00 |
| | | | Mission Impossible | 0.67 | 0.00 | 0.00 |
| | | | Othello | 1.00 | 0.33 | 0.00 |
| | | | Reasoning or Reciting | 0.40 | 0.00 | 0.00 |
| | | | Re-reading | 1.00 | 0.33 | 0.00 |
| | | | Tree of Thoughts | 0.83 | 0.00 | 0.00 |
| | | | VariErr-NLI | 1.00 | 0.33 | 0.00 |
| | | | WinoDict | 0.17 | 0.00 | 0.00 |
| | | Detailed Hints | CheckEval | 1.00 | 0.00 | 0.00 |
| | | | COGS | 0.50 | 0.33 | 0.00 |
| | | | Entity Tracking | 0.67 | 0.67 | 0.00 |
| | | | Explain then Translate | 1.00 | 1.00 | 0.67 |
| | | | Instruction Tuning | 0.67 | 0.00 | 0.00 |
| | | | Mission Impossible | 1.00 | 0.00 | 0.00 |
| | | | Othello | 1.00 | 0.33 | 0.33 |
| | | | Reasoning or Reciting | 0.27 | 0.00 | 0.00 |
| | | | Re-reading | 1.00 | 0.33 | 0.00 |
| | | | Tree of Thoughts | 0.67 | 0.00 | 0.00 |
| | | | VariErr-NLI | 1.00 | 0.00 | 0.00 |
| | | | WinoDict | 0.25 | 0.33 | 0.00 |

Table 22: Licenses for each Github repository.

| Task | Repository | License |
|------|-----------|---------|
| CheckEval | yukyunglee/CheckEval | MIT |
| COGS | najoungkim/COGS | MIT |
| Entity Tracking | najoungkim/code-models-entity-tracking | Apache 2.0 |
| Explain then Translate | PootieT/explain-then-translate | MIT |
| Instruction Tuning | john-hewitt/implicit-ins | Apache 2.0 |
| Mission Impossible | jkallini/mission-impossible-language-models | ??? |
| Othello | likenneth/othello_world | MIT |
| Reasoning or Reciting | ZhaofengWu/counterfactual-evaluation | MIT |
| Re-reading | EleutherAI/lm-evaluation-harness | MIT |
| Tree of Thoughts | princeton-nlp/tree-of-thought-llm | MIT |
| VariErr-NLI | mainlp/VariErr-NLI | MIT |
| WinoDict | google-research/language/tree/master/language/wino_dict | Apache 2.0 |

