# OpenReview forum: "RExBench: Can coding agents autonomously implement AI research extensions?"
_ICLR.cc/2026/Conference — ICLR 2026 Conference Withdrawn Submission_

### Official Review · Reviewer_E5Qr · 2025-10-17

**Soundness:** 2
**Presentation:** 3
**Contribution:** 1
**Rating:** 2
**Confidence:** 4

**Summary:**

This paper aims to benchmark the capabilities of LLM agents for modification-based coding implementation with specific focus on NLP and ML. Specifically, this paper proposes REXBench, which takes the original paper, the corresponding codebase, an instruction as the inputs, and then asks the agents to modify the codebase to implement the instruction. Automated evaluation metrics are proposed, i.e., by checking whether the results fall within a reasonable range.

**Strengths:**

- The writing is of good quality.
- Several LLMs and LLM agents are discussed in empirical results.

**Weaknesses:**

- From my perspective, this would not be an impactful benchmark for the community, due to several shortcomings: 1) limited number of tasks; 2) computational requirements, considering 8/12 tasks require A100; 3) the evaluation metric is unreliable: the agent would be rewarded as long as the execution results fall within a preset range of values, which is a quite loose evaluation metric.
- As this paper only proposes 12 tasks. It would be better the authors can provide all the task descriptions of REXBench.

**Questions:**

From my perspective, the proposed benchmark is of limited significance, and cannot benefit the future research of the community.

---

> ### Author Response · Authors · 2025-11-18
>
> While it is part of the peer review process to judge a paper’s significance, this tends to be one of the more subjective areas of assessment. However, in this case, we do actually have several pieces of anecdotal evidence that our work has significant value for the community. Without going into too much details to preserve anonymity, we can report the following indicators that our work is of interest to the community:
> * Multiple frontier model companies reached out to us expressing interest about it, mentioning the possibility of internal use.
> * Multiple maintainers of impactful collections of benchmarks (e.g., TerminalBench, Holistic Agent Leaderboard) are interested in integrating our benchmark into their evaluation pipelines.
>
> We therefore think that our benchmark CAN benefit the future research of the community and that there is, in fact, active interest.
>
> Regarding the computational requirements: We provide the evaluation infrastructure and cover the cost from our end, so we do not think that these requirements will be prohibitive for anyone to evaluate their agents on our tasks. Furthermore, we believe that it is important to consider ambitious evaluation scenarios, and in the domain of AI research, this generally entails the use of GPU resources.
>
> Regarding the tasks: A short description of all tasks is in the paper (see  Table 1 in Appendix A), and all task instructions are included in the Supplementary Material of the submission.

---

> > ### Comment · Reviewer_E5Qr · 2025-11-20
> >
> > Thanks for your response.
> >
> > - The authors do not address my concerns regarding 1) limited number of tasks; 2) the evaluation metric is unreliable: the agent would be rewarded as long as the execution results fall within a preset range of values, which is a quite loose evaluation metric.
> >
> > - I am also surprised to know that the authors plan to provide the evaluation infrastructure and cover the costs themselves. If this is indeed the case, I strongly recommend highlighting this point clearly in the paper, as it would substantially strengthen the contribution.

---

> > > ### Author Response · Authors · 2025-11-20
> > >
> > > The paper already states the following regarding evaluation infrastructure, which we believe sufficiently conveys that we will provide the evaluation infrastructure:
> > >
> > > > We will host this infrastructure using our own resources, and conduct evaluation asynchronously at a regular interval to update the leaderboard with the submissions we receive, similarly to Jimenez et al. (2024).
> > >
> > > but we can revise our description further to make this contribution stand out more. Thanks for the suggestion!
> > >
> > > Regarding limited number of tasks, we've included the following in the response to reviewer 6w6V:
> > >
> > > >We agree, however, that it would be good to have more tasks from more domains as part of the benchmark. As mentioned in the paper, we also see this work as a blueprint for how to perform such evaluations. While we cannot share details to not breach anonymity, we have been contacted by multiple researchers who are interested in adding tasks to the benchmark since we published a preprint of this work.
> > >
> > > Considering this interest in extending the benchmark, we expect the number of tasks to grow substantially in the future. Furthermore, given the massive performance differences between different agents at the moment, we already get a lot of signal about LLM and agent capabilities from what may seem like a small number of tasks. Note that many of the error bars in the results are not overlapping.
> > >
> > > Regarding agents gaming the evaluation: The allowed ranges of results are very narrow (sometimes when numbers can be replicated exactly across runs, we only allow one specific number that has several digits after the decimal point), so we consider it very unlikely that any solutions other than a true solution could achieve comparable results. Furthermore, we do require submitters to also submit traces of their agents so that it can be manually verified whether the agent solved the task or not and suspicious submissions can be filtered out, should this ever become necessary. We will clarify this point in the revision of the draft.

---

> > > > ### Comment · Reviewer_E5Qr · 2025-11-26
> > > >
> > > > Thanks for the response.
> > > > - I suggest highlighting the infrastructure resources in Introduction or even in Abstract.
> > > >
> > > > - For the limited number of the tasks, I choose to keep my opinion.
> > > >
> > > > - For the evaluation metric, I cannot be convinced by the statement “It is very unlikely that any solutions other than a true solution could achieve comparable results.” due to the high randomness of deep learning experiments.
> > > >
> > > > Considering these points, I choose to raise my score to 4 as a borderline reject.

---

### Official Review · Reviewer_YsPx · 2025-10-29

**Soundness:** 3
**Presentation:** 3
**Contribution:** 2
**Rating:** 2
**Confidence:** 4

**Summary:**

This paper introduces RExBench, a benchmark designed to evaluate the capability of LLM-based coding agents to autonomously implement research extensions in AI/NLP/ML. The benchmark consists of 12 tasks based on recently published papers, where agents must implement specified extensions starting from an existing codebase. The authors evaluate 13 agent configurations across three frameworks (aider, Claude Code, OpenHands) with various LLM backbones. Results show that even the best agents (OpenHands + Claude 4 Sonnet/GPT-5) achieve only ~31% success rate without hints, reaching ~47% with detailed hints. The paper provides extensive error analysis and recommendations for future agent development.

**Strengths:**

1. Research extension is a critical capability for autonomous research agents, distinct from replication
2. Novel implementations stored privately is a major strength over benchmarks like PaperBench
3. 13 agent configurations, 3 hint levels, detailed error taxonomy
4. VM-based evaluation with controlled execution environments ensures reproducibility
5. Distinction between explicit/implicit errors, over-editing observations, overthinking issues
6. Both short-term (scratchpads, repair mechanisms) and long-term (verification, context handling)

**Weaknesses:**

1. The work lacks a human baseline. It would be insightful to see how PhD students or domain experts perform on the same tasks. Similarly, there are no simpler non-agentic baselines to show what traditional systems could achieve. There’s also no direct comparison between agent-generated and human-written code, especially in terms of readability, maintainability, or long-term usability — all of which matter for research automation.

2. The experimental design has several weaknesses. There’s no statistical significance testing, even though the decoding process involves randomness. Inter-annotator agreement on the gold solutions isn’t reported, so we don’t know how consistent the labeling process was. The hint design also feels somewhat ad hoc — it’s unclear how the three hint levels were calibrated. Finally, the temperature setting differs between models, with some using 0.7 and others using defaults, which introduces inconsistency in evaluation.

3. Methodologically, the benchmark framework follows a fairly standard design. The main novelty lies in applying it to “research extension” tasks, rather than in introducing new benchmarking techniques. While the infrastructure is robust and well-documented, it doesn’t really bring conceptual innovation. The work is strong in execution but not particularly creative in method.

4. The authors themselves note that their instructions are much clearer and more informative than what a real researcher would encounter. This makes the setup somewhat artificial. Real research usually involves a lot of ambiguity, trial and error, and iterative exploration — all of which are missing here. The tasks also seem too narrowly defined and well-scoped, failing to capture the open-ended, exploratory nature of genuine research projects.

5. Some findings are left unexplained. For example, why do hints sometimes *reduce* performance, as seen in Othello and Tree-of-Thoughts tasks? There’s little analysis connecting specific task features to failure modes, and minimal discussion on when these agents should or shouldn’t be applied. The regression analysis in Figure 5 also seems underpowered, given that it’s based on only twelve data points. These gaps limit the depth of insight.

**Questions:**

1. How do PhD students perform on these tasks? What's the time/success rate comparison?
2. Did author measure inter-annotator agreement on gold implementations? Could multiple valid solutions exist?
3. How were the two hint levels designed? Was there any user study or pilot testing?
4. What percentage of failed attempts were close to success (within 1-2 bugs)? Could authors characterize the "distance to success"?
5.  Were there cases where agents found valid alternative implementations that didn't match authorsr gold numerical output but were scientifically sound?
6. Can authors provide objective difficulty metrics beyond lines of code (e.g., cyclomatic complexity, semantic changes required)?
7. The observation that reasoning models "overthink" is interesting—did authors try adjusting their reasoning effort parameters?
8. The authors mention agents make "unrequested modifications"—could instruction tuning on "minimal edits" help?
9. What's the cost-effectiveness compared to hiring a research assistant?

---

> ### Author Response · Authors · 2025-11-18
>
> Thank you very much for engaging so thoroughly with our paper, and your overall positive ratings for soundness, presentation, and contribution. We are also glad that many strengths of our work was recognized.
>
> Thank you also for the thoughtful clarifying questions, which we answer below.
>
> > How do PhD students perform on these tasks? What's the time/success rate comparison?
>
> We agree that a comparison between the speed and accuracy of PhD-level students and LLM agents would be an exciting extension for this work. However, we consider this follow-up work that falls outside of the scope of the current paper, and work as such would require thoughtful experiments and discussions that would merit a full paper of its own.
>
> > Did author measure inter-annotator agreement on gold implementations? Could multiple valid solutions exist?
>
> We assume that this question is about whether multiple gold implementations for each task exist and if so, whether all gold solutions lead to the same results, since typical IAA metrics  like kappa would not work for our solutions. Given that it takes PhD-level students on average one to two weeks to implement a gold implementation for an extension, we only have one gold implementation per task. However, the instructions, code, and results for each task were thoroughly reviewed by multiple team members, going through several rounds of iterations. This was to make sure that the instructions do not contain ambiguities that would allow for implementations that would lead to different results.
>
> > How were the two hint levels designed? Was there any user study or pilot testing?
>
> Given that this is a benchmark to assess LLM agents, we did not perform any user study in designing the hints (hints that would be beneficial to humans may not necessarily be beneficial to LLMs). The design was nevertheless still principled, because we each level on already known properties of LLMs: (1) they often find pinpointing useful information in long-context (“needle-in-a-haystack”) difficult, and (2) they benefit from spelled-out, step-by-step reasoning about the problem they are solving.
>
> For first-level hints, we provided information on which files need to be edited or are otherwise relevant, and/or which existing functions were critical for the extension. This was intended to help the agent localize the files and within them the relevant code that had to be modified to correctly implement the extension. In practice as well, we do observe failure based on localization errors: e.g., agents that tried to solve the task using just the baseline instructions sometimes failed to identify the correct files to edit (see also the right panel of Figure 2).
>
> For the second level of hints, we aimed to test whether agents can reliably implement the extension if they are provided with step-by-step instructions for one possible gold implementation that leads to the correct results. The second level of hints therefore provided the individual steps necessary (e.g., which functions need to be modified or implemented, which files need to be read, which library functions should be used) to implement the extension.
>
> (See also Section 4.2 of the paper.)
>
> We deliberately did not include any of the instructions with hints in the paper because the second level of hints provides such a detailed recipe on what the solutions should look like and we therefore want to avoid that this information leaks into the pretraining data of any future models. If you would like to verify one or two of the tasks with hints, we can find a way to share this information without risking data contamination. Please let us know if you would like to do this.
>
> > What percentage of failed attempts were close to success (within 1-2 bugs)? Could authors characterize the "distance to success"?
>
> This information for the best performing two models is in the paper - please refer to the subsection “Implicit Errors” in Section 5 for such an analysis.
>
> > Were there cases where agents found valid alternative implementations that didn't match authorsr gold numerical output but were scientifically sound?
>
> We manually analyzed the full output traces of the two best-performing agents (OpenHands + {Claude 4, GPT-5}), including the intermediate reasoning processes, and did not find any such instances. When the results deviated from the expected results, there were also critical bugs or logical flaws in the implementation.
>
> (continued...)

---

> > ### Author Response · Authors · 2025-11-18
> >
> > > Can authors provide objective difficulty metrics beyond lines of code (e.g., cyclomatic complexity, semantic changes required)?
> >
> > We’d be happy to include cyclomatic complexity in the final version of the paper, though one caveat there will be that some of the edits do not target procedural code but rather configuration files or bash scripts, so this metric may underestimate the complexity of the required changes. We are not aware of methods that directly measure the number of semantic changes (other than LLM-as-a-judge which we believe would not be an appropriate method given the limitations of the frontier models on our task) so we would appreciate pointers towards that if you are aware of methods that can compute this metric. We would be happy to update our analyses based on those.
> >
> > > The observation that reasoning models "overthink" is interesting—did authors try adjusting their reasoning effort parameters?
> >
> > Due to cost constraints, we did not investigate this systematically but we agree that this would be an interesting follow-up question to consider. Thanks for the suggestion!
> >
> > > The authors mention agents make "unrequested modifications"—could instruction tuning on "minimal edits" help?
> >
> > This is a very interesting question, and we believe that a benchmark like ours will be very useful in answering such questions regarding model tuning in the future in an empirically grounded way.
> >
> > > What's the cost-effectiveness compared to hiring a research assistant?
> >
> > Individual model runs cost at most USD 5 for one task but for many agent-model-task combinations, costs were below USD 1. While we did not track the time that PhD-level students took to implement, we estimate LLM agents to be two to three orders of magnitude cheaper than human research assistants. The time our PhD-and-beyond experts took to implement the gold solutions were quite variable across the tasks, some easier ones taking only 1-2 hours (in case they were already familiar with the original codebase) and harder ones, up to three weeks. However, we do not believe that cost effectiveness is a useful metric here, given that the reliability of the agent solutions was low—in research scenarios, the implementations not being high precision is quite high stakes.
> >
> > Additionally, we had a few questions and comments regarding some of the weaknesses you mentioned:
> >
> > > “Finally, the temperature setting differs between models, with some using 0.7 and others using defaults, which introduces inconsistency in evaluation.”
> >
> > As we mentioned in the paper, we used the default (unknown) setting only for frontier models that do not provide a method for setting this parameter. In other words, it is impossible to set this parameter for those models. While this may introduce a small inconsistency, we believe including models like GPT 5 and GPT o4-mini is nevertheless highly informative and does not diminish our claims—we would not want to exclude evaluating models of frontier capability because of this reason.
> >
> > **Regarding the perceived lack of conceptual innovation**: Among the strengths of the paper, you rightfully pointed out that our setup of using research extensions  is distinct from replication setups and that creating novel tasks and solutions is a strength over setups like PaperBench. We believe this is in fact a supporting argument towards the novelty in our benchmark—it does truly measure an important capability that existing benchmarks do not. However, if we missed work that has done something similar, please let us know and we’d be happy to discuss how/whether our approach differs from existing work.

---

> > > ### Author Response · Authors · 2025-11-18
> > >
> > > > The authors themselves note that their instructions are much clearer and more informative than what a real researcher would encounter. This makes the setup somewhat artificial. Real research usually involves a lot of ambiguity, trial and error, and iterative exploration — all of which are missing here. The tasks also seem too narrowly defined and well-scoped, failing to capture the open-ended, exploratory nature of genuine research projects.
> > >
> > > The instructions are indeed more informative than a one or two sentence summary that a real research assistant may encounter but, as discussed in the paper, we were aiming to find a good compromise between having open-ended research tasks and allowing for automatic evaluations. To make this task more open-ended, we would have to give up the idea of a benchmark setup, since the only way to reliably evaluate truly open-ended tasks would be through a one-time human evaluations (e.g., as done by Si et al., 2025; https://arxiv.org/abs/2506.20803). However, given the success of benchmarks like SWE-bench, which are also not as open-ended as everyday software engineering tasks, we consider the benchmark setup still highly useful to track high-level progress.
> > >
> > > Furthermore, the clearer and more informative setup likely leads to an overestimation of model abilities. Given that we find that all models fail at correctly implementing the extension most of the time, our results would likely be even more pronounced in a setup with even less guidance, so even though the setup may not be fully naturalistic, we still learn a lot from the behaviour of agents on our dataset.
> > >
> > > > “Some findings are left unexplained. For example, why do hints sometimes reduce performance, as seen in Othello and Tree-of-Thoughts tasks”
> > >
> > > We omitted details on these failure cases due to space constraints but we’d be happy to include them in the final version.
> > >
> > > For Othello, the more detailed instructions included reference to a helper function  (that already existed in the base repository) while the less detailed instructions did not make this reference. In the more detailed instructions setting, the agent incorrectly called the helper function, leading to an unsuccessful run. In the less detailed case, the agent re-implemented the behavior of the helper function as part of its solution and was able to use its own implementation successfully. This is why more hints reduced performance here.
> > >
> > > For the Tree-of-thoughts task, we observed something similar. With the first level of hints, none of the resulting patches used a helper function whereas all runs with the more detailed hints (which mentioned the helper function) called a specific helper function but did so incorrectly. This again led to a drop in performance.
> > >
> > > > “The regression analysis in Figure 5 also seems underpowered, given that it’s based on only twelve data points. These gaps limit the depth of insight.”
> > >
> > > This analysis is based on 36 datapoints (there are three runs each) and in the meantime, we have also replicated this analysis with 60 datapoints (with 5 runs each). If you would like more details about the new, higher powered analysis, we’d be happy to provide them.

---

> > > > ### Comment · Reviewer_YsPx · 2025-11-27
> > > >
> > > > I appreciate the very detailed and thoughtful rebuttal and I apologize that I should have expressed my question in more detail, but the author understood and answered my scattered questions in great depth. Many of my clarifying questions have been addressed: I now better understand the principled design of the two hint levels, the constraints that make inter-annotator agreement and multiple gold implementations impractical, and the reasons why hints can sometimes hurt performance (like incorrect reliance on helper functions).
> > > >
> > > > The authors also clarified that an analysis of distance to success for top models is already included (implicit errors), that no scientifically sound but numerically divergent solutions were observed in manual trace inspection, and that the regression analysis is based on more data points than I had realized.
> > > >
> > > > I think that the proposed addition of cyclomatic complexity as a difficulty metric is promising and agree that more open-ended, human-like setups and human baselines are valuable directions for following works, even if they are beyond the scope of this paper.
> > > >
> > > > While my concerns about the lack of human baseline and limits to naturalism remain to some extent, the rebuttal meaningfully strengthens my understanding of the contribution and I am ok to raise my score to 4.

---

### Official Review · Reviewer_6w6V · 2025-10-31

**Soundness:** 2
**Presentation:** 2
**Contribution:** 2
**Rating:** 2
**Confidence:** 3

**Summary:**

The paper proposes a benchmark REXBench which is designed to evaluate whether LLM agents can implement research extensions—i.e., modifications or follow-up experiments extending existing ML/NLP research papers. The benchmark includes 12 real research papers with corresponding codebases and expert-written instructions describing realistic extensions (e.g., changing models, datasets, or algorithms).  The experimental results show that current agents are still short of being able to handle realistic research extension tasks.

**Strengths:**

The paper proposes a novel problem formulation that tests agents’ ability to extend scientific research, an important while underexplored problem. The benchmark is well designed, including containerized evaluation, de-contamination, etc. The experiments are thorough, using various evaluation metrics and also including cost/time study and an error analysis.

**Weaknesses:**

Although the problem is novel, it is somewhat too niche. In addition, the scale and scope of the benchmark are very limited — it contains only 12 papers and covers only the NLP/ML domain. Therefore, it does not sufficiently evaluate the model’s capability in research extension.

**Questions:**

See weaknesses

---

> ### Author Response · Authors · 2025-11-18
>
> Thank you for taking the time to review our paper!
>
> Regarding the scope of the benchmark, we do think that the problem of implementing research code is not niche, in fact, it's very common. Likely 90%+ of the thousands of ICLR submissions this year required some form of research code implementation, and we believe that rigorously assessing to what extent LLM-generated code in this domain leads to correct results seems like a pressing and sufficiently broad-scope question given the increased use of LLM agents by AI/ML researchers. And while there was a bias in the selection of tasks given the expertise of our team in ML/NLP, we still believe that ML/NLP is scope-wise quite broad, and we include a wide range of subdomains (model, evaluation, algorithm, data) as shown in Table 1 in the Appendix. Our takeaways (see Conclusion) are also quite general and not restricted to our specific domain: we would be very surprised if the results were specific to our tasks, and consequently, LLM-based coding agents could perfectly implement research extensions in other domains that we did not include in the benchmark, or they showed completely different patterns of failure in other domains.
>
> We agree, however, that it would be good to have more tasks from more domains as part of the benchmark. As mentioned in the paper, we also see this work as a blueprint for how to perform such evaluations. While we cannot share details to not breach anonymity, we have been contacted by multiple researchers who are interested in adding tasks to the benchmark since we published a preprint of this work.

---

### Official Review · Reviewer_DkXJ · 2025-11-01

**Soundness:** 3
**Presentation:** 3
**Contribution:** 3
**Rating:** 6
**Confidence:** 4

**Summary:**

This paper propsoes a new benchmark RExBench which can evaluate the code agents for incremental research ideas.The benchmark includes 12 research papers and their corresponding extensions. Compared to previous tasks, the aim is to automatically assess how well an agent can autonomously implement realistic research extensions.

**Strengths:**

1. The aim of benchmarking is to automatically assess how well an agent can autonomously implement realistic research extensions. This goal is kind of realistic and interesting. The extension proposal is annotated with the gold edits for the target extension, which is a good resource.
2. The paper evaluates 13 LLM agents based on this benchmark. The additional hints setting serves as an ablation study for the bottleneck of the pipeline.  The paper also includes both quantitative and qualitative analysis.
3. The paper includes code. The paper has good visualizations. The paper includes most of the infrastructure pipelines in the main context.

**Weaknesses:**

1. Some details of the benchmark are missing. What is the average extension for each of those 12 papers? The title is kind of misleading. Instead of research extension, the paper is more on the adaptation of the existing code base.
2. Although errors are discussed in section 5.1, most of them are pretty high-level. The paper also fails to explain the reason behind those errors. The reason behind those errors can help researchers understand the drawbacks of current methods. Sec 5.2 is kind of high-level. If they can be linked to the specific errors, it would be better.
3. The conclusion seems a bit too long. The paper might need to include a small subset for human evaluation to quantify some of the observations. If section 5 can be supported by more numbers or evidence, it will become stronger.

**Questions:**

See weakness

---

> ### Author Response · Authors · 2025-11-18
>
> Thank you for taking the time to review our paper, we are grateful for your positive evaluation of our work.
>
> Regarding the weaknesses that you identified, we had a couple of follow-up questions that would help us address them. Specifically:
>
> - Could you clarify what you mean by  “average extension”? We do provide a high-level categorization for the different extensions in Table 1 in Appendix A but perhaps you had a specific metric such as “average number of lines of code changed in the extension” in mind here? We’d be happy to provide this information if this is indeed what you intended, since we already have these metrics computed in order to conduct the statistical analysis we provided in the paper.
> - Could you elaborate a bit on why “adaptation of the existing code base” does not count as research extension when the original codebase implements the research project? In our view, extending an existing research codebase so that it is possible to run new experiments is a core part of what constitutes a research extension in the domain of empirical ML/AI research but let us know if you think this is still misleading or incorrect.
> - You pointed out that the paper “fails to explain the reason behind those errors. The reason behind those errors can help researchers understand the drawbacks of current methods.” We  point out that we do provide possible explanations behind some of the error types: e.g., aider’s single-turn structure being limited in long context tasks as ours, lack of a scratchpad that could catch minor errors, etc., and this is also reflected in our explicit discussion of the drawbacks of current methods in the conclusion when making recommendations for future agent design. But we agree that the explanations can in theory be deeper. However, unfortunately, we are not aware of any analysis methods that would allow for causal analyses of frontier models for which very little information on their architecture, training, and data is available, and for which we cannot obtain model activations. At the same time, we found that models with more information available, such as Deepseek, are generally not able to even follow the overall agent instructions, so deeper analyses of more open models, likely would not translate to the reasons for errors in frontier models. However, if you had any suggestions for methods we can adopt to perform a deeper analysis, we’d very much appreciate it, and would be happy to implement them  in the final version of the paper.
> - Regarding your point around the conclusion: “If section 5 can be supported by more numbers or evidence, it will become stronger”. Did you have any specific numbers or evidence in mind that you would like to see? We’d be happy to include more analyses.
> - Similarly, regarding the human evaluation: Which observations did you have in mind here?
>
> Other points:
> - Regarding the long conclusion: We’ll aim to streamline this more for the final version of the paper, thanks for the suggestion!

---

> > ### Comment · Reviewer_DkXJ · 2025-11-26
> >
> > Thanks for your explanation. I am still very skeptical about the problem setting and the dataset size. Additionally, the paper fails to include any open-source model for detailed error analysis. Therefore, I keep my score.

---

### Note · Authors · 2025-12-15

I have read and agree with the venue's withdrawal policy on behalf of myself and my co-authors.